# Quantifying $N_2O$ reduction to $N_2$ based on $N_2O$ isotopocules – validation with independent methods (helium incubation and [15]N gas flux method)

Dominika Lewicka-Szczebak[1*], Jürgen Augustin[2], Anette Giesemann[1], Reinhard Well[1]

[1]Thünen Institute of Climate-Smart Agriculture, Federal Research Institute for Rural Areas, Forestry and Fisheries, Bundesallee 50, D-38116 Braunschweig, Germany;
[2]Leibniz Centre for Agricultural Landscape Research, Eberswalder Straße 84, D-15374 Müncheberg, Germany;

*Correspondence to*: Dominika Lewicka-Szczebak (dominika.lewicka@thuenen.de)

**Abstract.** Stable isotopic analyses of soil-emitted $N_2O$ ($\delta^{15}N^{bulk}$, $\delta^{18}O$ and $\delta^{15}N^{sp}$ = [15]N site preference within the linear $N_2O$ molecule) may help to quantify $N_2O$ reduction to $N_2$, an important but rarely quantified process in the soil nitrogen cycle. The $N_2O$ residual fraction (remaining unreduced $N_2O$, $r_{N2O}$) can be theoretically calculated from the measured isotopic enrichment of the residual $N_2O$. However, various $N_2O$ producing pathways may also influence the $N_2O$ isotopic signatures, and hence complicate the application of this isotopic fractionation approach.

Here this approach was tested based on laboratory soil incubations with two different soil types applying two reference methods for quantification of $r_{N2O}$: helium incubation with direct measurement of $N_2$ flux and the [15]N gas flux method. This allowed a comparison of the measured $r_{N2O}$ values with the ones calculated based on isotopic enrichment of residual $N_2O$. The results indicate that the performance of the $N_2O$ isotopic fractionation approach is related with the accompanying $N_2O$ and $N_2$ source processes and the most critical is the determination of the initial isotopic signature of $N_2O$ before reduction ($\delta_0$). We show that $\delta_0$ can be well experimentally determined if stable in time and then successfully applied for determination of $r_{N2O}$ based on $\delta^{15}N^{sp}$ values. Much more problematic is to deal with temporal changes of $\delta_0$ values leading to failure of the approach based on $\delta^{15}N^{sp}$ values only. For this case we propose here a dual $N_2O$ isotopocule mapping approach, where calculations are based on the relation between $\delta^{18}O$ and $\delta^{15}N^{sp}$ values. This allows for the simultaneous estimation of the $N_2O$ producing pathways contribution and the $r_{N2O}$ value.

# 1 Introduction

$N_2O$ reduction to $N_2$ is the last step of microbial denitrification, i.e., anoxic reduction of nitrate ($NO_3^-$) to $N_2$ through the following intermediates: $NO_3^- \rightarrow NO_2^- \rightarrow NO \rightarrow N_2O \rightarrow N_2$ (Firestone and Davidson, 1989; Knowles, 1982). Commonly applied analytical techniques enable us to quantitatively analyse only the intermediate product of this process, $N_2O$, but not the final product, $N_2$. This is due to the high atmospheric $N_2$ background precluding direct measurements of $N_2$ emissions (Bouwman et al., 2013; Saggar et al., 2013). Hence, $N_2O$ reduction to $N_2$ is the least well understood N transformation and constitutes a key quantity of the N cycle, as potential significant loss of reactive N to the atmopshere. $N_2$ and $N_2O$ denitrification fluxes cause lowering of both plant available N and N leaching and $N_2O$ reduction to $N_2$ decreases $N_2O$ fluxes (Butterbach-Bahl et al., 2013).

To overcome the problems with $N_2$ quantification, three methods for $N_2$-flux estimation are applicable (Groffman, 2012; Groffman et al., 2006): direct $N_2$-measurements under a $N_2$-free helium atmosphere (helium incubation method), [15]N analyses of gas fluxes after addition of [15]N-labelled substrate ([15]N gas flux method), and the reduction inhibition method based on the comparison of $N_2O$ fluxes with and without acetylene application (acetylene inhibition method). These methods were widely applied in laboratory studies to determine the contribution of $N_2O$ reduction to $N_2$, which is usually expressed as the fraction of the residual unreduced $N_2O$: $r_{N2O} = y_{N2O}/(y_{N2}+y_{N2O})$ ($y$: mole fraction). The whole scale of possible $r_{N2O}$ variations, ranging from 0 to 1, had been found in laboratory studies (Lewicka-Szczebak et al., 2015; Mathieu et al., 2006; Morse and Bernhardt, 2013; Senbayram et al., 2012). However, due to technical limitations, only the [15]N gas flux method can be applied under field conditions to determine the $r_{N2O}$ (Aulakh et al., 1991; Baily et al., 2012; Bergsma et al., 2001; Decock and Six, 2013; Kulkarni et al., 2013; Mosier et al., 1986). The acetylene inhibition method is not useful for field studies due to catalytic NO decomposition in presence of $C_2H_2$ and $O_2$ (Bollmann and Conrad, 1997; Felber et al., 2012; Nadeem et al., 2013) and the helium incubation method requires a sophisticated air-tight incubation system, so far attainable only in laboratory conditions. Hence, no comprehensive data sets from field-based measurements of soil $N_2$ emissions are available and this important component in soil nitrogen budget is still missing. This constitutes a serious shortcoming in understanding and mitigating the microbial consumption of nitrogen fertilisers (Bouwman et al., 2013;

Seitzinger, 2008), and the N$_2$O emission, which significantly contributes to global warming and stratospheric ozone depletion (IPCC, 2007; Ravishankara et al., 2009).

N$_2$O isotopic fractionation studies could potentially be used for quantification of $r_{N2O}$ under field conditions (Park et al., 2011; Toyoda et al., 2011; Zou et al., 2014). Its advantage over the [15]N gas flux method lies in its easier and non-invasive application, no need of additional fertilization, and much lower costs. This expands the application potential of the isotopic fractionation method and enables its more widespread use. This kind of study uses the isotopic analyses of the residual unreduced N$_2$O, of which three isotopic signatures can be determined: of oxygen ($\delta^{18}$O), bulk nitrogen ($\delta^{15}$N$^{bulk}$) and nitrogen site preference ($\delta^{15}$N$^{sp}$), i.e., the difference in $\delta^{15}$N between the central and the peripheral N atom of linear N$_2$O molecules (Brenninkmeijer and Röckmann, 1999; Toyoda and Yoshida, 1999). All these three isotopic signatures ($\delta^{18}$O, $\delta^{15}$N$^{bulk}$ and $\delta^{15}$N$^{sp}$) are altered during the N$_2$O reduction process and the magnitude of the observed change depends largely on the N$_2$O residual fraction (Jinuntuya-Nortman et al., 2008; Menyailo and Hungate, 2006; Ostrom et al., 2007; Well and Flessa, 2009a). Hence, principally, this fraction can be calculated from the isotopic enrichment of the residual N$_2$O, provided that the isotopic signature of the initially produced N$_2$O before reduction ($\delta_0$) and the net isotope effect associated with N$_2$O reduction ($\eta_{red}$) are known (Lewicka-Szczebak et al., 2014). $\delta_0^{15}$N and $\delta_0^{18}$O values depend largely on the isotopic signatures of the N$_2$O precursors, i.e., of NH$_4^+$, NO$_3^-$, NO$_2^-$, H$_2$O, and on the transformation pathways, e.g., nitrification or denitrification (Perez et al., 2006). $\delta_0^{15}$N$^{sp}$ values, however, are independent of the precursors, but differ according to different pathways, e.g., nitrification or denitrification (Sutka et al., 2006) and different microbial communities, e.g., bacterial or fungal denitrifiers (Rohe et al., 2014; Sutka et al., 2008) involved in the N$_2$O production. Therefore, $\delta_0$ values may vary between different soils and due to different conditions, e.g., moisture, temperature, fertilization. $\eta_{red}$ values are variable depending on experimental conditions, but these variations are largest for $\eta_{red}^{18}$O and $\eta_{red}^{15}$N$^{bulk}$, whereas for $\eta_{red}^{15}$N$^{sp}$ quite stable values in the range from -7.7 to -2.3 ‰ with an average of -5.4±1.6 ‰ have been found (Lewicka-Szczebak et al., 2014). Moreover, recently this value has been confirmed under oxic atmosphere (Lewicka-Szczebak et al., 2015), hence, it can be expected that $\delta^{15}$N$^{sp}$ values can be applied as a robust basis to calculate N$_2$O reduction also for field studies.

However, some open questions still remain: (i) are the isotopic fractionation factors for denitrification processes determined in laboratory experiments transferable to field conditions?; (ii) how robustly can the $N_2O$ residual fraction be determined?; (iii) is the quantification of the entire nitrogen loss due to denitrification possible? In this study we present a validation of the calculations based on the $N_2O$ isotopic fractionation performed in laboratory experiments. Two different reference methods for quantification of $N_2O$ reduction were applied: incubation in $N_2$-free helium atmosphere and the [15]N gas flux method. Helium incubations allow for simultaneous determination of the $N_2O$ isotopic signature and the $r_{N2O}$ from the same incubation vessel (Lewicka-Szczebak et al., 2015), whereas in [15]N gas flux experiments, parallel incubations of [15]N-labelled and natural abundance treatments are necessary. Nevertheless, [15]N-labelled treatments provide additional information on the coexisting $N_2O$-forming processes (Müller et al., 2014), which might possibly impact the $N_2O$ isotopic signatures. Therefore, here we have applied both methods for the same pair of very different soils, a mineral arable and an organic grassland soil, aiming at a better understanding of the complex $N_2O$ production and consumption in these soils. The main aims of this study were to (i) check how precisely the $N_2O$ residual fraction can be calculated with the isotopic fractionation approach, (ii) identify the sources of possible bias, e.g., coexisting $N_2O$ forming processes, and (iii) search for the possibilities to improve the precision and applicability of this calculation approach.

## 2 Methods

The list with explanations of all abbreviations and specific terms used in the manuscript can be found in the Supplement (S1).

### 2.1 Experimental set-ups

### 2.1.1 Experiment 1 - helium incubation as reference method (Exp1)

Two soil types were used: a mineral arable soil with silt loam texture classified as a *Haplic Luvisol* (Min soil) and an organic grassland soil classified as *Histic Gleysol* (Org soil). The soils were air dried and sieved at 4mm mesh size. Afterwards, the soil was rewetted to obtain 70 % water-filled pore space (WFPS) and fertilised with 50 mg N (added as $NO_3$) per kg soil. Then soils were thoroughly mixed to

obtain a homogenous distribution of water and fertilizer and 250 cm$^3$ of wet soil were repacked into each incubation vessel with bulk densities of 1.4 g cm$^{-3}$ for the Min soil and 0.4 g cm$^{-3}$ for the Org soil. Afterwards the water deficit to the target WFPS: 70 or 80 % WFPS depending on the treatment, was added on the top of the soil. The incubations were performed using a special gas-tight incubation system allowing for application of a N$_2$-free atmosphere. This system has been described in detail by Eickenscheidt et al. (2014). Here we present briefly its general idea.

The incubation vessels were cooled to 2 ºC, repeatedly evacuated (to 0.047 bar), flushed with He to reduce the N$_2$ background and afterwards flushed with a continuous stream of He+O$_2$ for at least 60 hours. When a stable and low N$_2$ background (below 10ppm) was reached, temperature was increased to 22 ºC. The incubation lasted 5 days, while the headspace was constantly flushed with a continuous flow of 20 % O$_2$ in helium (He/O$_2$) mixture for the first 3 days and then with pure He for the following 2 days, at a flow rate of ca. 15 cm$^3$ min$^{-1}$. The fluxes of N$_2$O and N$_2$ were directly analyzed and the samples for N$_2$O isotopocule analyses were collected at least twice a day. The N$_2$O residual fraction was determined based on the direct measurement of N$_2$O and N$_2$ fluxes.

The data from two selected samplings of this experiment have been already published with particular emphasis on the O isotopic fractionation (experiment 2.3-2.6 in (Lewicka-Szczebak et al., 2016)).

## 2.1.2 Experiment 2 – $^{15}$N gas flux as reference method (Exp2)

The same soils (Min soil and Org soil) as in Exp1 were used for parallel incubations under either an anoxic (N$_2$) or an oxic (78 % He + 2 % N$_2$ + 20 % O$_2$) atmosphere with continuous gas flow at 10 cm$^3$ min$^{-1}$. The N$_2$ background concentration in the oxic incubation was reduced to increase the sensitivity of the $^{15}$N gas flux method (Meyer et al., 2010).

The soils were air dried and sieved at 4mm mesh size. Afterwards, the soil was rewetted to obtain a WFPS of 70 % and fertilised with 80 mg N (added as NO$_3^-$) per kg soil. Half of each soil was fertilized with *Chile saltpeter* (NaNO$_3$, Chili Borium Plus, Prills-Natural origin, supplied by Yara, Dülmen, Germany), i.e., nitrate fertilizer from atmospheric deposition ore with δ$^{15}$N at natural abundance level (NA treatment). This fertilizer was used to enable determining O exchange between

denitrification intermediates and water based on the $^{17}O$ anomaly of *Chile saltpeter* (Lewicka-Szczebak et al., 2016). The other half of the soil was fertilized with $^{15}N$-labelled $NaNO_3$ (98 at% $^{15}N$) ($^{15}N$ treatment). Then soils were thoroughly mixed to obtain a homogenous distribution of water and added fertilizer. 500 $cm^3$ of wet soil was repacked into incubation vessels with bulk densities of 1.4 g $cm^{-3}$ for

the Min soil and 0.4 g $cm^{-3}$ for the Org soil. Afterwards the water deficit to the target WFPS of 75 % for Min soil and 85 % for Org soil was added on the top of the soils. Glass jars (0.8 $dm^3$ J. WECK GmbH u. Co. KG, Wehr, Germany) were used with airtight rubber seal and with two three-way valves installed in their glass cover to enable continuous gas flow and sampling. The sampling vials were connected to vents of the incubation jars (Well et al., 2008) and were exchanged each 24 h. The soils were incubated

for 9 days at constant temperature (22 ℃). During each sampling, gas samples were collected in two 12 $cm^3$ Labco Exetainers® (Labco Limited, Ceredigion, UK) and for NA treatment additionally in one 120 $cm^3$ crimped vial.

## 2.2 Chromatographic analyses

In Exp1, online trace gas concentration analysis of $N_2$ was performed with a micro-GC (Agilent

Technologies, 3000 Micro GC), equipped with a thermal conductivity detector (TCD). Concentrations of trace gases were analysed by a GC (Shimadzu, Duisburg, Germany, GC–14B) equipped with an electron capture detector (ECD) for $N_2O$ and $CO_2$. The measurements precision was better than 20 ppb for $N_2O$ and 200 ppb for $N_2$, respectively.

In Exp2 the samples for gas concentration analyses were collected in Labco Exetainer® (Labco

Limited, Ceredigion, UK) vials and were analysed using an Agilent 7890A gas chromatograph (Agilent Technologies, Santa Clara, CA, USA) equipped with an ECD detector. Precision as given by the standard deviation ($1\sigma$) of four standard gas mixtures was typically 1.5%.

## 2.3 Soil analyses

Soil water content was determined by weight loss after 24h drying in 110℃. Soil nitrates and

ammonium were extracted in 0.01 M $CaCl_2$ solution (1:10 ratio) by shaking at room temperature for one

hour and $NO_3^-$ and $NH_4^+$ concentrations were determined colorimetrically with an automated analyser (Skalar Analytical B.V., Breda, The Netherlands).

## 2.4 Isotopic analyses in NA treatments

### 2.4.1 Isotopic signatures of $N_2O$

Gas samples were analysed using an isotope ratio mass spectrometer (Delta V, Thermo Fisher Scientific, Bremen, Germany) coupled to an automatic preparation system (Precon + Trace GC Isolink, Thermo Fisher Scientific, Bremen, Germany) where $N_2O$ was pre-concentrated, separated and purified. In the mass spectrometer, $N_2O$ isotopocule values were determined by measuring $m/z$ 44, 45, and 46 of the intact $N_2O^+$ ions as well as $m/z$ 30 and 31 of $NO^+$ fragment ions. This allows the determination of average $\delta^{15}N$ ($\delta^{15}N^{bulk}$), $\delta^{15}N^{\alpha}$ ($\delta^{15}N$ of the central N position of the $N_2O$ molecule), and $\delta^{18}O$ (Toyoda and Yoshida, 1999). $\delta^{15}N^{\beta}$ ($\delta^{15}N$ of the peripheral N position of the $N_2O$ molecule) was calculated from $\delta^{15}N^{bulk} = (\delta^{15}N^{\alpha} + \delta^{15}N^{\beta}) / 2$ and $^{15}N$ site preference ($\delta^{15}N^{sp}$) from $\delta^{15}N^{sp} = \delta^{15}N^{\alpha} - \delta^{15}N^{\beta}$. The scrambling factor and $^{17}O$-correction were taken into account (Röckmann et al., 2003). Pure $N_2O$ (Westfalengas; purity $>$ 99.995 %) was used as internal reference gas. It had been analyzed for isotopocule values in the laboratory of the Tokyo Institute of Technology using calibration procedures reported previously (Toyoda and Yoshida, 1999; Westley et al., 2007). Moreover, the standards from a laboratory intercomparison (REF1, REF2) were used for performing two-point calibration for $\delta^{15}N^{sp}$ values (Mohn et al., 2014).

All isotopic values are expressed as ‰ deviation from the $^{15}N/^{14}N$ and $^{18}O/^{16}O$ ratios of the reference materials (i.e., atmospheric $N_2$ and Vienna Standard Mean Ocean Water (V-SMOW), respectively). The analytical precision determined as standard deviation ($1\sigma$) of the internal standards for measurements of $\delta^{15}N^{bulk}$, $\delta^{18}O$ and $\delta^{15}N^{sp}$ was typically 0.1, 0.1, and 0.5 ‰, respectively.

### 2.4.2 Isotopic signatures of $NO_3^-$

$\delta^{18}O$ and $\delta^{15}N$ of nitrate in the soil solution were determined using the bacterial denitrification method (Sigman et al., 2001). The analytical precision determined as standard deviation ($1\sigma$) of the international standards was typically 0.5 ‰ for $\delta^{18}O$ and 0.2 ‰ for $\delta^{15}N$.

### 2.4.3 Soil water analyses

Soil water was extracted with the method described by Königer et al. (2011) and $\delta^{18}O$ of water samples was measured using a cavity ring down spectrometer Picarro L1115-*i* (Picarro Inc., Santa Clara, USA). The analytical precision determined as standard deviation ($1\sigma$) of the internal standards was below 0.1 ‰. The overall error associated with the soil water extraction method determined as standard deviation ($1\sigma$) of the 5 samples replicates was below 0.5 ‰.

## 2.5 Isotopic analyses in $^{15}N$ treatments

### 2.5.1 $^{15}NO_3$ and $^{15}NH_4$

$^{15}N$ abundances of $NO_3^-$ ($a_{NO3-}$) and $NH_4^+$ ($a_{NH4+}$) were measured according to the procedure described in Stange et al. (2007). $NO_3^-$ was reduced to NO by Vanadium-III-chloride ($VCl_3$) and $NH_4^+$ was oxidized to $N_2$ by Hypobromide ($NaOBr$). NO and $N_2$ were used as measurement gas. Measurements were performed with a quadrupole mass spectrometer (GAM 200, InProcess, Bremen, Germany).

### 2.5.2 $^{15}N_2O$ and $^{15}N_2$

The gas samples from the $^{15}N$ treatments of Exp2 were analysed for *m/z* 28 ($^{14}N^{14}N$), 29 ($^{14}N^{15}N$) and 30 ($^{15}N^{15}N$) of $N_2$ using a modified GasBench II preparation system coupled to an isotope ratio mass spectrometer (MAT 253, Thermo Fisher Scientific, Bremen, Germany) according to Lewicka-Szczebak et al. (2013a). This system allows a simultaneous determination of isotope ratios $^{29}R$ ($^{29}N_2/^{28}N_2$) and $^{30}R$ ($^{30}N_2/^{28}N_2$) representing three separated gas species ($N_2$, $N_2+N_2O$ and $N_2O$), all measured as $N_2$ gas after $N_2O$ reduction in a Cu oven.

For each of the analysed gas species ($N_2$, $N_2+N_2O$ and $N_2O$) the fraction originating from the $^{15}N$-labelled pool ($f_P$) was calculated after Spott et al. (2006) as:

$$f_P = \frac{a_M - a_{bgd}}{a_P - a_{bgd}} \tag{1}$$

where:

$a_M$: $^{15}N$ abundance in total gas mixture

$$a_M = \frac{^{29}R + 2\ ^{30}R}{2(1 + ^{29}R + ^{30}R)} \tag{2}$$

$a_{bgd}$: $^{15}N$ abundance of non-labelled pool (atmospheric background or experimental matrix)

$a_P$: $^{15}N$ abundance of $^{15}N$-labelled pool, from which the $f_P$ was derived:

$$a_P = \frac{^{30}x_M - a_M \cdot a_{bgd}}{a_M - a_{bgd}} \tag{3}$$

The calculation of $a_P$ is based on the non-random distribution of $N_2$ and $N_2O$ isotopologues (Spott et al., 2006) where $^{30}x_M$ is the fraction of $^{30}N_2$ in the total gas mixture:

$$^{30}x_M = \frac{^{30}R}{1 + ^{29}R + ^{30}R} \tag{4}$$

Identical calculations are performed for each separated gas species providing the values $f_{P\_N2}$, $a_{P\_N2}$ and $f_{P\_N2O}$, $a_{P\_N2O}$ and $f_{P\_N2+N2O}$, $a_{P\_N2+N2O}$. Importantly, in our incubations under artificial atmosphere, we have no background $N_2O$, hence the $^{15}N$ abundance of total $N_2O$ ($a_{M\_N2O}$) results from the mass balance of the $^{15}N$ abundances and sizes of the pools contributing to $N_2O$ production. Because $a_{P\_N2O}$ represents the $^{15}N$ abundance of the $^{15}N$-labelled pool emitting $N_2O$, the $a_{M\_N2O}$ value enables to distinguish between $N_2O$ originating from labelled $^{15}NO_3^-$ pool ($f_{P\_N2O}$) and from non-labelled natural abundance pools, like $NH_4^+$ or organic N ($f_{N\_N2O}$), as:

$$a_{M\_N2O} = a_{P\_N2O} \cdot f_{P\_N2O} + 0.003663 \cdot f_{N\_N2O} \tag{5}$$

where 0.003663 is the fraction of $^{15}N$ in non-labelled $N_2O$ and $f_{N\_N2O} = 1 - f_{P\_N2O}$.

Based on the determined $f_{P\_N2}$ and $f_{P\_N2+N2O}$ we can calculate $r_{N2O}$ as:

$$r_{N2O} = \frac{y_{N2O}}{y_{N2} + y_{N2O}} = \frac{f_{P\_N2+N2O} - f_{P\_N2}}{f_{P\_N2+N2O}} \tag{6}$$

where $y$ represents the mole fractions. This approach appeared to be more suitable than directly using $f_{P\_N2O}$, because (i) direct isotopic analysis of the $N_2O$ was not possible in samples with low $N_2O$ concentration and (ii) $f_{P\_N2}$ and $f_{P\_N2+N2O}$ were quantified in one sample based on the same method whereas $f_{P\_N2O}$ includes analysis of isotope ratios of the $N_2O$ peak and analysis of $N_2O$ conentration by

gas chromatography in a replicate gas sample, thus resulting in potential bias in $f_{P\_N2O}$ due to the difficulty to collect exactly identical replicate gas samples.(Lewicka-Szczebak et al., 2013b).

Knowing $r_{N2O}$ we can estimate the total denitrification [$N_2+N_2O$] flux using the measured [$N_2O$] flux and the determined $r_{N2O}$ as:

$$5 \quad [N_2+N_2O]\,flux = \frac{[N_2O]\,flux \cdot f_{P\_N2O}}{r_{N2O}} + [N_2O]\,flux \cdot f_{N\_N2O} \tag{7}$$

Moreover, from the comparison of the $a_{P\_N2}$ or $a_{P\_N2O}$ with $a_{NO3-}$ values obtained from $NO_3^-$ analysis of soil extracts, the contribution of hybrid $N_2$ ($f_{H\_N2}$) and $N_2O$ ($f_{H\_N2O}$) can be estimated. If $a_P <$ $a_{NO3-}$ this can be due to the combination of two N sources, labelled and non-labelled, to form $N_2O$ or $N_2$ (Spott and Stange, 2011). Hence, the fractions of three pools: non-labelled ($N$), labelled non-hybrid ($L$) and labelled hybrid ($H$) contributing to $N_2$ or $N_2O$ formation were determined according to Spott and Stange (2011):

$$N = \frac{a_{NO3-}^{2} + a_{NO3-}(-2\,^{30}x - ^{29}x) + ^{30}x}{(a_{bgd} - a_{NO3-})^{2}} \tag{8}$$

$$L = \frac{a_{bgd}^{2} + a_{bgd}(-2\,^{30}x - ^{29}x) + ^{30}x}{(a_{bgd} - a_{NO3-})^{2}} \tag{9}$$

$$H = \frac{a_{bgd}(2\,^{30}x + ^{29}x - 2\,a_{NO3-}) + a_{NO3-}(2\,^{30}x + ^{29}x) - 2\,^{30}x}{(a_{bgd} - a_{NO3-})^{2}} \tag{10}$$

15 and the hybrid fraction, for either $N_2O$ or $N_2$, is calculated as:

$$f_H = \frac{H}{L+H} \tag{11}$$

and:

$$f_L + f_H = 1 \tag{12}$$

## 2.6 Co-existence of other N-transformation processes

20 The mineral N concentrations and $^{15}N$ abundances allow for a quantification of:

(i)     formation of natural abundance $NO_3^-$ via gross nitrification ($n$) based on the dilution of the $^{15}$N-labelled $NO_3^-$ pool, which is obtained from the initial (subscript $0$) and final (subscript $t$) concentration ($c$) and $^{15}$N abundance ($a$) in soil nitrate (Davidson et al., 1991):

$$n = (c_{NO3\_0} - c_{NO3\_t}) \cdot \frac{\log(a_{NO3\_0}/a_{NO3\_t})}{\log(c_{NO3\_0}/c_{NO3\_t})} \tag{13}$$

5   (ii)    formation of $^{15}$N-labelled $NH_4^+$, most probably due to *DNRA* (dissimilatory nitrate reduction to ammonium) or due to coupled immobilisation-mineralisation (Rutting et al., 2011), based on $^{15}$N mass balance of final (subscript $t$) and initial (subscript $0$) ammonium concentration ($c$) and $^{15}$N abundance ($a$) in final and initial ammonium and average (of initial and final value, subscript $av$) $^{15}$N abundance in nitrate :

$$10 \quad DNRA = \frac{c_{NH4\_t} \cdot a_{NH4\_t} - c_{NH4\_0} \cdot a_{NH4\_0}}{a_{NO3\_av}} \tag{14}$$

(iii)   mineralisation ($m$) - amount of natural abundance N which was added to the system, based on N balance, including final and initial ammonium concentration ($c_{NH4\_t}$, $c_{NH4\_0}$), nitrification ($n$), non-labelled $N_2O$ flux ($f_{N\_N2O}$*[$N_2O$] flux) and *DNRA*:

$$m = c_{NH4\_t} - c_{NH4\_0} + n + f_{N\_N_2O} \cdot [N_2O] \text{flux} - DNRA \tag{15}$$

15   (iv)   nitrate immobilisation ($i$) - magnitude of N sink not explained by other processes, including final and initial nitrate concentration ($c_{NO3\_t}$, $c_{NO3\_0}$), nitrification ($n$), total N-gas flux [$N_2O+N_2$] flux and *DNRA*:

$$i = c_{NO3\_0} - c_{NO3\_t} + n - DNRA - [N_2O + N_2] \text{flux} \tag{16}$$

20   **2.7 $N_2O$ isotopic fractionation to quantify $N_2O$ reduction**

The $N_2O$ fractionation approach is based on the changes in $N_2O$ isotopic signatures due to partial $N_2O$ reduction to $N_2$, which alters the $\delta^{18}O$, $\delta^{15}N^{bulk}$ and $\delta^{15}N^{sp}$ of the residual unreduced $N_2O$ ($\delta_r$). All these isotopic signatures depend on the $N_2O$ residual fraction ($r_{N2O}$) according to the following isotopic fractionation equations applying closed system Rayleigh model (Mariotti et al., 1981):

$$\frac{1+\delta_r}{1+\delta_0} = (r_{N_2O})^{\eta_{red}} \qquad (17)$$

or in simplified, approximated form (applied only for graphical interpretations in Sect. 3.4.1):

$$\delta_r \approx \delta_0 + \eta_{red} \cdot \ln(r_{N_2O}) \qquad (18)$$

To be able to determine $r_{N2O}$ from $N_2O$ isotopic values of individual samples according to Eq. (17), isotopic fractionation factors associated with $N_2O$ reduction ($\eta_{red}$) and initial $N_2O$ isotopic signature before reduction ($\delta_0$) must be known. We tested various experimental approaches to determine $\eta_{red}$ and $\delta_0$ values to check which value yields best fit between calculated and measured $N_2O$ reduction and thus to identify, which of the methods to determine $\eta_{red}$ and $\delta_0$ is the most suitable one.

### 2.7.1 Estimating $\eta_{red}$ and $\delta_0$ values

*Mean $\eta_{red}$ and $\delta_0$ values for the entire experiment*

From the statistically significant logarithmic fits between $r_{N2O}$ and measured $\delta_r$ values we can estimate the isotopic fractionation by $N_2O$ production ($\delta_0$) and $N_2O$ reduction ($\eta_{red}$) according to Eq. (18), where the slope represents the $\eta_{red}$, the isotope effect associated with $N_2O$ reduction, and the intercept gives $\delta_0$, the initial isotopic signature for the produced $N_2O$ unaffected by its reduction (Fig. 4)

For $\delta^{18}O$ and $\delta^{15}N^{bulk}$ , $\delta_0$ values are expressed as relative values in relation to the source, i.e., soil water ($\delta^{18}O(N_2O/H_2O)$) and soil nitrate ($\delta^{15}N^{bulk}(N_2O/NO_3)$). This allows us to reasonably compare different treatments differing in soil water isotopic signatures and properly interpret $\delta^{15}N^{bulk}$ values which are related to the isotopic signature of nitrate, getting enriched with incubation time. $\delta_0^{15}N^{sp}$ is independent of the isotopic signature of the source, hence the measured $\delta^{15}N^{sp}$ values were directly used for determination of correlations.

*Temporarily changing $\eta_{red}$ and $\delta_0$ values*

The interpretations and calculations based on $\delta$ values are difficult when we deal with the simultaneous variations in $r_{N2O}$ and $\delta_0$ values. Usually, to calculate $r_{N2O}$ a stable $\delta_0$ is assumed (Lewicka-Szczebak et al., 2015) and to precisely determine temporal changes in $\delta_0$, we need independent data on $r_{N2O}$ (Köster et al., 2015). In field studies both $r_{N2O}$ and $\delta_0$ cannot be determined precisely, but rather the possible

ranges for each parameter can be given (Zou et al., 2014). In our experiments we have measured $r_{N2O}$ with independent methods, hence we can assess the $\delta_0$ changes with time, under the assumption that $\eta_{red}$ is stable, or conversely, assess changes in $\eta_{red}$ assuming stable $\delta_0$ values. The assumption of a stable $\eta_{red}$ value is best justified for $\eta_{red}{}^{15}N^{sp}$, which shows the narrowest range of variations from -7.7 to -2.3 ‰

5  with a mean of -5 ‰ (Lewicka-Szczebak et al., 2015; Lewicka-Szczebak et al., 2014). Hence, a fixed $\eta_{red}{}^{15}N^{sp}$ value of -5 ‰ was used to calculate a $\delta_0{}^{15}N^{sp}$ value for each sample and thus to estimate its change with time. To calculate the possible temporal change in $\eta_{red}$ values, $\delta_0$ was assumed constant. The respective $\delta_0$ value derived from the correlation between $\ln(r_{N2O})$ and $\delta_r$ (Mariotti et al., 1981) was used.

*Fungal fraction estimated from $\delta_0$ values*

From the calculated $\delta_0{}^{15}N^{sp}$ values, the fraction of N$_2$O originating from fungal denitrification ($f_F$) can be estimated using the isotopic mass balance. Isotopic endmembers for $\delta^{15}N^{sp}$ values were assumed to be 35 ‰ for fungal denitrification (Rohe et al., 2014) and -5 ‰ for heterotrophic bacterial

denitrification (Sutka et al., 2006; Toyoda et al., 2005). The mixing endmember characterized by higher $\delta^{15}N^{sp}$ values can theoretically also originate from nitrification (hydroxylamine oxidation pathway), but only in the oxic treatments. However, in our experimental set-up, due to high nitrate amendment, no ammonia amendment and high soil moisture, N$_2$O flux from nitrification should be much lower than from denitrification (Zhu et al., 2013). Therefore, the significant shifts in $\delta_0{}^{15}N^{sp}$ values observed here

are rather discussed as a result of fungal denitrification admixture.

**2.7.2 Calibration and validation of $r_{N2O}$ quantification**

The precision of the quantification of the N$_2$O reduction based on the N$_2$O isotopic fractionation approach was checked by comparison of the calculated values and the values measured by the reference

methods, i.e. direct N$_2$ measurements in He incubation (for Exp1) and [15]N gas flux method (for Exp2). The $\delta_0$ and $\eta_{red}$ values needed to determine $r_{N2O}$ with Eq. (18) were found from the ln fit between the

isotopic signature of residual unreduced $N_2O$ and $r_{N2O}$ determined by the independent method, as shown in the previous section 2.7.1.

The calibration of the isotopic fractionation approach was performed by applying $\delta_0{}^{15}N^{sp}$ and $\eta_{red}{}^{15}N^{sp}$ values obtained in the particular experiment to calculate $r_{N2O}$ from the same experiment. The precision of this approach was evaluated by comparing measured and calculated $r_{N2O}$ and determining the standard error of calculated $r_{N2O}$.

The validation of the isotopic fractionation approach was performed by applying $\delta_0{}^{15}N^{sp}$ and $\eta_{red}{}^{15}N^{sp}$ values determined in a parallel experiment to calculate $r_{N2O}$ of the validation experiment with the same soil. The validation was performed in three ways (Val1 – Val3):

(i)     Val1 used $\delta_0{}^{15}N^{sp}$ and $\eta_{red}{}^{15}N^{sp}$ values obtained from a previous static experiment performed with the same soil (Exp 1E-F in Lewicka-Szczebak et al. (2014)) to calculate $r_{N2O}$ for Exp1 and 2 based on the measured $\delta{}^{15}N^{sp}$ values of residual unreduced $N_2O$.

(ii)    Val2 used $\delta_0{}^{15}N^{sp}$ and $\eta_{red}{}^{15}N^{sp}$ values obtained from Exp1 to calculate $r_{N2O}$ for Exp2, and *vice versa*.

(iii)   Val3 used the same $\delta_0{}^{15}N^{sp}$ as Val2, but for $\eta_{red}{}^{15}N^{sp}$ the common value of -5 ‰ was applied, as recently suggested as a mean robust $\eta_{red}{}^{15}N^{sp}$ (Lewicka-Szczebak et al., 2014). Here we checked how our results are affected when we use this common value instead of the $\eta_{red}{}^{15}N^{sp}$ value determined for the particular soil.

## 2.7.3 Mapping approach to distinguish mixing and fractionation processes

Until now, isotopomer "maps", i.e. plots of $\delta{}^{15}N^{sp}$ vs $\delta{}^{15}N^{bulk}$ or $\delta{}^{15}N^{sp}$ vs $\delta{}^{18}O$, have been used to differentiate between processes (Koba et al. (2009), Zou et al. (2014)) or to identify $N_2O$ reduction to $N_2$ (Well et al., 2012). Here we present a very first attempt of simultaneous quantification of fractionation and mixing processes based on the relation between $\delta{}^{15}N^{sp}$ and $\delta{}^{18}O$ values, which we call 'mapping approach'. The graphical illustration of the $\delta{}^{15}N^{sp}/\delta{}^{18}O$ "maps" is presented in Fig. 1. The approach is based on the different slopes of the mixing line between bacterial denitrification and fungal denitrification or nitrification and the reduction line reflecting isotopic enrichment of residual $N_2O$ due

to its partial reduction. Both lines are defined from the known most relevant literature data on the respective $\delta_0$ and $\eta_{red}$ values:

- $\delta_0{}^{15}N^{sp}$ *from pure culture studies for bacterial denitrification*: for heterotrophic bacterial denitrification from -7.5 to +3.7 ‰ (Sutka et al., 2006; Toyoda et al., 2005) and for nitrifier denitrification from -13.6 to +1.9 ‰ (Frame and Casciotti, 2010; Sutka et al., 2006). As both processes overlap, a common mean endmember value for $N_2O$ production by bacterial denitrification of -3.9 ‰ is used.

- $\delta_0{}^{18}O(N_2O/H_2O)$ *for bacterial denitrification*: for heterotrophic bacterial denitrification from controlled soil incubations from 17.4 to 21.4 ‰ (Lewicka-Szczebak et al., 2016; Lewicka-Szczebak et al., 2014) and for nitrifier denitrification based on pure culture studies from 19.8 to 26.5 ‰ (Frame and Casciotti, 2010; Sutka et al., 2006). As both processes overlap, a common endmember value for $N_2O$ production by bacterial denitrification of 21 ‰ is used. (For heterotrophic bacterial denitrification we used the values of the controlled soil incubation only (from 17.4 to 21.4 ‰) and disregarded pure culture studies which show a large range of possible values due to various O-exchange with ambient water depending on the bacterial strain, whereas soil incubations indicated that this exchange is high (Kool et al., 2007; Snider et al., 2013) and the isotope effect between water and formed $N_2O$ quite stable (Lewicka-Szczebak et al., 2016).)

- $\delta_0{}^{15}N^{sp}$ *for fungal denitrification and nitrification based on pure culture studies*: for fungal denitrification from 30.2 to 39.3 ‰ (Maeda et al., 2015; Rohe et al., 2014; Sutka et al., 2008) and for nitrification from 32.0 to 38.7 ‰ (Frame and Casciotti, 2010; Heil et al., 2014; Sutka et al., 2006). As both processes overlap, a common endmember value for $N_2O$ production by fungal denitrification of 34.8 ‰ is used. (A recent study indicated also a lower $\delta_0{}^{15}N^{sp}$ value for one individual fungal species, which was disregarded here due to its very low $N_2O$ production: *C. funicola* showed $\delta_0{}^{15}N^{sp}$ of 21.9 ‰ but less than 100 times lower $N_2O$ production with nitrite compared to other species, and no $N_2O$ production with nitrate (Rohe et al., 2014). Similarly, from the study of Maeda et al. (2015) we accepted only the values of strains with higher $N_2O$ production (> 10mg $N_2O$-N/g biomass).)

- $\delta_0{}^{18}O(N_2O/H_2O)$ *for fungal denitrification and nitrification based on pure culture studies:* for fungal denitrification from 40.6 to 51.9 ‰ (Maeda et al., 2015; Rohe et al., 2014; Sutka et al., 2008) and for nitrification from 35.6 to 55.2 ‰ (Frame and Casciotti, 2010; Heil et al., 2014; Sutka et al., 2006). As both processes overlap, a common endmember value for $N_2O$ production by fungal denitrification of 43.6 ‰ is used. (The relevant values for fungal denitrification are selected after the same criteria as above for $\delta_0{}^{15}N^{sp}$.)

- *Isotopic fractionation factors associated with $N_2O$ reduction:* values obtained from controlled soil incubations are $\eta_{red}{}^{15}N^{sp}$ from -7.7 to -2.3 ‰ with a mean of -5 ‰ and of $\eta_{red}{}^{18}O$ values from -25 to -5 ‰ with a mean of -15 ‰ (Jinuntuya-Nortman et al., 2008; Lewicka-Szczebak et al., 2014; Menyailo and Hungate, 2006; Ostrom et al., 2007; Well and Flessa, 2009a). Although the range of possible $\eta_{red}$ variations is quite large, it has been shown recently that the mean values and typical $\eta_{red}{}^{15}N^{sp}/ \eta_{red}{}^{18}O$ ratios are applicable for oxic or anoxic conditions unless $N_2O$ reduction is almost complete, i.e. $r_{N2O} < 0.1$ (Lewicka-Szczebak et al., 2015).

The $\delta^{15}N^{sp}/\delta^{18}O$ slope of the mixing line between the endmember value for $N_2O$ production of fungal denitrification / nitrification and heterotrophic bacterial denitrification / nitrifier denitrification is distinct from the slope of the reduction line resulting from reduction isotope effects (Fig. 1: reduction line and mixing line, respectively). Isotopic values of the samples analyzed are typically located between these two, reduction and mixing, lines. From their position on the $\delta^{15}N^{sp}/\delta^{18}O$ "map" we can estimate the impact of fractionation associated with $N_2O$ reduction and admixture of $N_2O$ originating from fungal denitrification / nitrification. If we assume bacterial denitrification as the first source of $N_2O$, then we can deal with two scenarios:

(i) Scenario 1 (Sc1): the $N_2O$ emitted due to bacterial denitrification is first reduced (point move along reduction line up to the intercept with red_mix line) and then mixed with the second endmember (point move along red_mix line to the measured sample point)

(ii) Scenario 2 (Sc2): the $N_2O$ from two endmembers is first mixed (point move along mixing line up to the intercept with mix_red line) and only afterwards the mixed $N_2O$ is reduced (point move along mix_red line to the measured sample point).

While both scenario yield identical results for the admixture of $N_2O$ from fungal denitrification / nitrification, the resulting reduction shift, and hence the calculated $r_{N2O}$ value, is higher when using Sc2.

## 3 Results

### 3.1 Exp1,

*$N_2O$ and $N_2$ fluxes and isotopocules of $N_2O$*

The detailed results presented as time series are shown in the supplement Fig. S1. In general, the switch from oxic to anoxic conditions resulted in an increase of gaseous N-losses. For both treatments of the Min soil (70 and 80 % WFPS), we observed a gradual decrease in $r_{N2O}$ with incubation time, from 1 down to 0.25 for 80 % WFPS and down to 0.63 for 70 % WFPS. This is associated with a simultaneous increase in $\delta$ values, from 21.6 to 59.1 ‰ for $\delta^{18}O$, from -52.9 to -29.9 ‰ for $\delta^{15}N^{bulk}$, and from 0.3 to 19.6 ‰ for $\delta^{15}N^{sp}$. For the Org soil 80 % WFPS treatment, the initial increase in $r_{N2O}$, from 0.08 to 0.49 during the oxic phase, is followed by a slight drop (from 0.60 to 0.39) during the anoxic phase. $\delta$ values did not show a clear trend over time and ranged from 11.2 to 41.9 ‰ for $\delta^{18}O$, from -46.4 to -17.4 ‰ for $\delta^{15}N^{bulk}$ and from -1.9 to 17.5 ‰ for $\delta^{15}N^{sp}$. In the 70 % WFPS treatment, the gas fluxes were below detection limit during the oxic phase.

$\delta^{18}O(H_2O)$ of soil water ranged from -6.5 to -5.1 ‰ for Org and Min soil, respectively.

### 3.2 Exp2

### 3.2.1 NA treatment, Exp2

*$N_2O$ and $N_2$ fluxes and isotopocules of $N_2O$*

The detailed results presented as time series are shown in the supplement Fig. S2. For the anoxic treatments we observe a gradual decrease in $N_2O$ flux and an increase in $N_2$ flux (calculated with the $r_{N2O}$ values determined in the parallel $^{15}N$ treatment) with incubation progress. For Min soil, $\delta^{18}O$ increases from 27.3 to 71.2 ‰, $\delta^{15}N^{bulk}$ from -45.6 to -28.2 ‰, and $\delta^{15}N^{sp}$ from 5.5 to 34.6 ‰. For Org soil $\delta^{18}O$ increases from 18.4 to 52.6 ‰, $\delta^{15}N^{bulk}$ from -46.2 to +7.5 ‰, and $\delta^{15}N^{sp}$ from 4.3 to 31.4 ‰.

Under oxic conditions, we observe much higher standard deviations for both $N_2O$ flux and $N_2O$ isotopic signatures. For Min soil no clear trend over time can be described: the $N_2O$ flux is decreasing but rises again at the end of the incubation. Similarly, $\delta$ values first increase and then decrease again varying between 32.8 and 63.4 ‰ for $\delta^{18}O$, between -43.2 and -3.0 ‰ for $\delta^{15}N^{bulk}$ and between 3.1 and 16.8 ‰ for $\delta^{15}N^{sp}$ (Fig. S2.2(a)). For Org soil, $\delta$ values increase until the 5$^{th}$ day, from 17.5 to 46.6 ‰ for $\delta^{18}O$ and from -48.4 to -38.1 ‰ for $\delta^{15}N^{bulk}$, and then vary around 46 and -39 ‰, respectively. $\delta^{15}N^{sp}$ values keep increasing through the entire incubation period from 1.7 to 23.6 ‰ (Fig. S2.2(b)).

$\delta^{18}O(H_2O)$ of soil water ranged from -8.5 to -6.1 ‰ for Org and Min soil, respectively.

### 3.2.2 $^{15}N$ treatment, Exp2

*$N_2O$ and $N_2$ fluxes and $^{15}N$ enrichment of N pools*

The detailed results presented as time series are shown in the supplement Fig. S3. The determined $r_{N2O}$ values in the anoxic treatments are decreasing with incubation progress, from 0.58 to 0.02 for Min soil (Fig. S3.1(a)) and from 0.71 to 0.30 for Org soil (Fig. S3.1(b)). In the oxic treatments $r_{N2O}$ varies between 0.08 and 0.72. The minimum values are reached about in the middle of the incubation time in both soil types: on the 6$^{th}$ day for Min soil and the 5$^{th}$ day for Org soil incubation.

From all $^{15}N$ treatments only for the anoxic Org soil treatment provided very consistent $^{15}N$ atom fractions in all gaseous fractions ($a_{M\_N2O}$, $a_{P\_N2O}$, $a_{P\_N2}$). They ranged from 42 to 46 at%, which is in close agreement with soil nitrate ($a_{NO3}$=43 at%) (Fig. S3.1(b)). For the anoxic Min soil treatment, $a_{P\_N2}$ and $a_{P\_N2O}$ ranged from 49 to 51 at% and also correspond to $a_{NO3}$ (51 at%), but the $^{15}N$ atom fraction of the emitted $N_2O$ ($a_{M\_N2O}$) is significantly lower, decreasing from 49 to 24 at% with incubation time (Fig. S3.1(a)). In oxic conditions we deal with even lower $^{15}N$ atom fractions in total $N_2O$. $a_{M\_N2O}$ ranges from 4 to 32 at% for Min soil (Fig. S3.2(a)) and from 11 to 37 at% for Org soil (Fig. S3.2(b)). Moreover, for oxic treatments also lower values of $a_{P\_N2}$ can be observed, down to 28 at% for Min soil and 34 at% for Org soil. For mineral N we observed almost no change in $^{15}N$ content in the extracted nitrate under anoxic conditions, with maximal change in $a_{NO3}$ of 0.3 at%. Under oxic conditions a slight decrease of 1.5 at% for Min and 3.2 at% for Org soil occurs. The non-labelled ammonium pool stays mostly unchanged under oxic treatments, but significant $^{15}N$ enrichment is observed under anoxic

conditions, where $a_{NH4}$ reaches 8.7 at% for Min and 3.5 at% for Org soil by the end of the incubation (Fig. S3.1(a), S3.1(b)).

*N transformations*

In Table 1, calculated rates of N transformations are shown. Initial and final concentrations for nitrate and ammonium were measured, total gaseous N-loss ($[N_2+N_2O]$ flux) is calculated (Eq. (7)), the rates of nitrification (*n*), *DNRA*, mineralisation (*m*), immobilisation (*i*) were estimated according to Eqs. (13) - (16). The flux of $N_2O$ from non-labelled soil N pools was calculated as $f_{N\_N2O} \times [N_2O]$ flux. The nitrification rate (*n*) was highest for the Org soil in oxic conditions (1.93 mg N per kg soil and 24 h). But even in anoxic treatments, a low *n* rate was detected (up to 0.06 mg N). In the anoxic treatments *DNRA* was also active, which resulted in formation of $^{15}N$ labelled $NH_4^+$ (from 0.02 to 0.10 mg N, for Min soil and Org soil, respectively). Mineralisation (*m*) appears to be very high for Org soil, both in oxic (1.99 mg N) and anoxic (1.25 mg N) conditions, and lower for Min soil (0.31 and 0.15 mg N, respectively). Interestingly, in each treatment a quite pronounced additional nitrate sink, most probably due to N immobilisation (*i*), was found, mostly much larger than the total gaseous loss ($[N_2+N_2O]$ flux) (Table 1).

*$N_2O$ and $N_2$ source processes*

Based on the non-random distribution of $N_2O$ isotopologues obtained in $^{15}N$ treatments, we can differentiate between the $^{15}N$-pool derived $N_2O$ ($f_{P\_N2O}$) and non-labelled $N_2O$ fraction ($f_{N\_N2O}$) (Fig. 2). $f_{P\_N2O}$ decreases with lowering of total $N_2O$ fluxes and is higher for anoxic treatments (above 0.42 for Min soil and above 0.91 for Org soil) when compared to oxic treatments (from 0.03 to 0.67 and from 0.14 to 0.98, respectively). A significant contribution of non-labelled $N_2O$ ($f_{P\_N2O} < 1$) in the anoxic Min soil treatment was thus evident (Fig. 2(a)), but the lower $f_{P\_N2O}$ values are associated with lower $N_2O$ fluxes at the end of the incubation, and the cumulative flux of non-labelled $N_2O$ is only approx. 0.02 of the total denitrification flux $[N_2O+N_2]$. This is slightly higher than for the Org soil anoxic treatment, where the cumulative flux of non-labelled $N_2O$ reaches only ca. 0.01 of the total denitrification flux $[N_2O+N_2]$. The contribution of the cumulative non-labelled $N_2O$ flux to the total denitrification flux

[N₂O+N₂] is quite significant for oxic treatments, with a mean value of 0.18 and 0.29 for Org soil and Min soil, respectively. Within the $^{15}$N-pool derived N₂O, the hybrid sub-fraction can be determined ($f_{H\_N2O}$). Hybrid N₂O was found only in oxic treatments (Fig. 2). For Min soil, $f_{H\_N2O}$ was detected in all measured N₂O samples and varied between 0.05 and 0.19. For Org soil, no $f_{H\_N2O}$ was found during the first two or three days of incubation when the N₂O concentration was highest. Afterwards its contribution gradually increased with decreasing N₂O concentration, reaching up to 0.25 of the $^{15}$N-pool derived N₂O. Similarly, $f_{H\_N2}$ was determined. Very small $f_{H\_N2}$ was detected in anoxic treatments, up to 0.09 for Min soil and up to 0.18 for Org soil, where only five samples from two vessels indicated possible presence of hybrid N₂ (Fig. 3). Significantly higher $f_{H\_N2}$ were observed for oxic conditions, up to 0.90 for Min soil and up to 0.68 for Org soil. For Org soil, there is significant negative correlation between $f_H$ and, both, N₂O (Fig.2) and N₂ flux (Fig.3), whereas no such relation exists for Min soil.

### 3.3 N₂O isotopic fractionation to quantify N₂O reduction

### 3.3.1 Estimating $\eta_{red}$ and $\delta_0$ values

For Min soil we obtained very consistent correlations between $r_{N2O}$ and measured $\delta_r$ values for all treatments except the oxic Exp2. The N₂O fluxes for oxic conditions showed large variations within the repetitions and between the treatments (compare Fig. S2.2(a) and S3.2(a)) which indicates that NA and $^{15}$N treatment are not directly comparable. Therefore, the results of the oxic incubation (blue diamonds, Fig. 4(a)) show no correlation between $\delta^{15}N^{sp}$ and $r_{N2O}$. The other three fits indicate an absolutely consistent value for $\delta_0^{15}N^{sp}$ from 4.0 to 4.5 ‰ and also a quite consistent value for $\eta_{red}^{15}N^{sp}$ from -8.6 to -6.7 ‰ (Fig. 4(a)). Much wider ranges of $\eta_{red}$ values were found for $\eta_{red}^{18}O$ (from -22.7 to -9.9 ‰) and $\eta_{red}N^{bulk}$ (from -6.6 to -2.0 ‰). In contrast to quite variable $\eta_{red}$ values, the determined $\delta_0$ values are very robust, with $\delta_0^{18}O$ about +36 and $\delta_0^{15}N^{bulk}$ about -45 ‰ (Table 2).

These relations look very different for Org soil. Firstly, there is no significant correlation between $\delta_r$ and $r_{N2O}$ for Exp1, whereas all correlations are significant for Exp2 (Fig. 4(b), Table 2). The $\eta_{red}$ values determined for Exp2 for Org soil (Table 2) are much more negative than for Min soil and

also compared to the known literature range of fractionation factors (Jinuntuya-Nortman et al., 2008; Lewicka-Szczebak et al., 2015; Well and Flessa, 2009a).

*Temporarily changing $\eta_{red}$ and $\delta_0$ values*

Theoretical $\delta_0{}^{15}N^{sp}$ values were calculated for individual samples assuming stable $\eta_{red}$ values (as described in Sect. 2.7.1) and the variations of calculated $\delta_0{}^{15}N^{sp}$ with incubation time for both soils are presented in Fig. 5. An increase in $\delta_0{}^{15}N^{sp}$ value with time is observed for both soils, but is much larger and clearly unidirectional for Org soil. Since $r_{N2O}$ simultaneously decreases during the incubation, the $\delta_0{}^{15}N^{sp}$ value obtained from the correlation between $\delta^{15}N^{sp}$ and $r_{N2O}$ (Table 2, Fig. 4(b)) is much below the actual one (Fig. 5(b)). For Min soil this increasing trend is not so large and constant, and hence the correlation between $\delta^{15}N^{sp}$ and $r_{N2O}$ (Table 2, Fig. 4(a)) provides the $\delta_0{}^{15}N^{sp}$ value which represents the mean of actual variations quite well (Fig. 5(a)).

It could also be assumed that $\delta_0$ values are constant during the experiment and the variable $\eta$ values can be calculated. Under this assumption the $\eta$ values through both soils and experiments are extremely variable for $\eta^{15}N^{bulk}$ from -59 to +30 ‰, for $\eta^{15}N^{sp}$ from -24 to +15 ‰, and for $\eta^{18}O$ from -143 to +48 ‰.

*Fungal fraction estimated from $\delta_0$ values*

For Org soil, the time course of $\delta_0{}^{15}N^{sp}$ values (Fig. 5) indicated a very pronounced increase in the fraction of N₂O originating from fungal denitrification ($f_F$) during the incubation time of Exp2 (9 days), giving $f_F$ values from 10 % at the beginning up to 75 % at the end. For Min soil in Exp2, $f_F$ was smaller and varied from 7 to 49 %.

### 3.3.2 Calibration and validation of $r_{N2O}$ quantification

From the correlation tested above (Table 2) we found that only for Min soil $\delta_0$ and $\eta_{red}$ values can be robustly determined from $\delta^{15}N^{sp}$ values. Hence, we show here the calibration and validation based on these values only. The calibration shows a quite good agreement between the measured and the calculated $r_{N2O}$ with a significant fit to the 1:1 line (Fig. 6). The mean absolute difference between

measured and calculated $r_{N2O}$ was 0.08 for Exp1 and 0.04 for Exp2. The mean relative error in the determination of the reduced $N_2O$ fraction (1- $r_{N2O}$) representing the $N_2$ flux was 36 % for Exp1 and 8 % for Exp2. For Exp1 we have tested if a better fit could be obtained when fractionation factors for oxic and anoxic treatment are determined and applied separately. In Fig. 6, points calculated with mean values for oxic and anoxic treatment (Exp1 mean) as well as calculations for either oxic or anoxic treatments are shown. The fit to a 1:1 line is similar for the calculation using the mean values (Exp1 mean: $R^2$=0.83) and the respective oxic and anoxic treatments considered individually (Exp1 oxic: $R^2$=0.86 and Exp1 anoxic: $R^2$=0.79).This indicates that for this soil $\eta_{red}$ values were not affected by incubation conditions.

For Val1, i.e. using the $\delta_0{}^{15}N^{sp}$ and $\eta_{red}{}^{15}N^{sp}$ values obtained from a previous static experiment performed with the same soil, the calculated and measured values showed a correlation but the observed slope was significantly lower than 1 (Fig. 7 (red triangles)). For Exp1 the mean absolute difference between the measured and the calculated $r_{N2O}$ reaches 0.41 and the relative error in determining $N_2$ flux is as high as 234 %, whereas for Exp2 these values are much lower with 0.09 and 16 %, respectively. Significantly lower errors determined for Exp2 are due to many data points of extremely low $r_{N2O}$ values.

For Val2, i.e. using $\delta_0{}^{15}N^{sp}$ and $\eta_{red}{}^{15}N^{sp}$ values from Exp1, the fit to the 1:1 line was definitely much better than for Val1, which is shown by the significant correlation between measured and calculated $r_{N2O}$ (Fig. 7 (black triangles)). The absolute mean difference between the measured and the calculated $r_{N2O}$ was 0.10 and 0.07 for Exp1 and Exp2, and the relative error in determining the $N_2$ flux reached 54 % and 13 %, respectively. Nevertheless, for Exp2 the maximal difference of 0.40 is very high. The four samples showing the highest deviation are the very first samples of the incubation, which most probably show slightly different microbial activity compared to the further part of the incubation. As shown in Fig. 5, at the beginning we deal with larger dominance of bacterial over fungal $N_2O$, which results in lower $\delta_0{}^{15}N^{sp}$ than assumed in the calculations, and consequently in an overestimation of the $r_{N2O}$.

For Val3, i.e. using a common value of -5 ‰ for $\eta_{red}{}^{15}N^{sp}$, the fit is very similar as for Val2 (not shown). For Exp1 the mean absolute difference between measured and calculated $r_{N2O}$ was 0.14

(relative error 60 %), which was slightly higher compared to the 0.10 difference (relative error 54 %) for Val2. For Exp2 this difference was only 0.05 (relative error 9%), hence even lower than 0.07 (relative error 13 %) obtained for Val2.

Summarising the results of these three validation scenarios, we can conclude that actual $\delta_0$ values must apparently be known to obtain reliable estimates of $r_{N2O}$, whereas it seems possible to use a general value for $\eta_{red}{}^{15}N^{sp}$.

### 3.3.3 Mapping approach to distinguish mixing and fractionation processes

As a qualitative indicator of mixing and fractionation processes we analysed relations between pairs of isotopic signatures to determine the slopes for the measured $\delta$ values. The same was done for the $\delta_0$ values calculated using the measured $r_{N2O}$ values (Eq. (17)). All the calculated slopes are presented in Table 3, and graphical illustrations are shown in the supplement (Fig. S4). The $\delta^{15}N^{sp}/\delta^{18}O$ slopes for Org soil are generally higher (from 0.65 to 0.76) than for Min soil (from 0.30 to 0.64) (Table 3). But we can also notice that for both soils, the slopes in Exp1 are lower than in Exp2 The slopes between $\delta^{18}O/\delta^{15}N^{bulk}$ observed in our study range mostly from 1.94 to 3.25 (Table 3). Only for Org soil in anoxic conditions (in both Exp1 and Exp2) this slope is substantially lower from 0.61 to 0.84.

With the mapping approach we used dual isotope values, i.e. $\delta^{15}N^{sp}$ and $\delta^{18}O$, to calculate $r_{N2O}$ and the fraction of N$_2$O originating from fungal denitrification or nitrification ($f_F$) as described in Sect. 2.7.3. This was done for both soils but with Exp2 data only (Fig. 8). Both scenarios provide identical results for $f_F$ values, whereas $r_{N2O}$ values are always higher for Sc2 ("first reduction, then mixing") when compared to Sc1 ("first mixing, then reduction") with maximal difference up to 0.39 between them. Figure 8 shows the comparison between calculated and measured $r_{N2O}$ values. For most results the measured value is within the range of values obtained from both scenarios. For Org soil, Sc2 results show better agreement with the measured values, but rather the opposite is observed for the Min soil. The oxic treatment for Min soil shows the worst agreement with the measured values, i.e., the calculated values indicate pronounced underestimation of $r_{N2O}$. The calculated $f_F$ values exhibit a continuous increase with incubation time for all treatments except the oxic treatment of Min soil.

## 4. Discussion

### 4.1 N$_2$O and N$_2$ source processes

In this study quite a high contribution of non-labelled N$_2$O was documented (Fig. 2, Fig. 3). Non-labelled N$_2$O may originate from nitrification or nitrifier denitrification (Wrage et al., 2001). However, in the conditions favouring denitrification with high soil moisture (WFPS 75 %) the typical N$_2$O yield from nitrification is much lower compared to the N$_2$O yield from denitrification (Butterbach-Bahl et al., 2013; Well et al., 2008). Therefore, in these experimental conditions the contribution of nitrification to N$_2$O fluxes should be rather negligible. Most surprising is the significant contribution of non-labelled N$_2$O ($f_{P\_N2O} < 1$) in the anoxic Min soil treatment associated with lower N$_2$O fluxes at the end of incubation (Fig. 2(a)). Moreover, for both soils in the anoxic treatment the cumulative non-labelled N$_2$O flux in mg N is higher than the initial NH$_4^+$ pool plus the NH$_4^+$ possibly added due to *DNRA* (Table S1). This indicates that oxidation of organic N must be active in these treatments. Recently, it has been shown that this process can be even the dominant N$_2$O producing pathway (Müller et al., 2014); however, it is questionable if this can be active also under anoxic conditions. Nitrifier denitrification or eventually also some abiotic N$_2$O production would be the most probable processes to produce non-labelled N$_2$O in anoxic treatments, but since the substrate is NH$_4^+$, it must have been preceded by ammonification of organic N.

A higher contribution of non-labelled N$_2$O was noted for oxic treatments (Fig. 2). This flux can be well explained by nitrification, because it represents, respectively, 2 and 3 % of the nitrification rate (Table 1), which is at the upper end of the known range for the nitrification product ratio (Well et al., 2008). Nitrification was quite significant in oxic treatments and NO$_3^-$ production from nitrification exceeded largely the NH$_4^+$ available at the beginning of the incubation (Table S1). This indicated that a pronounced amount of organic N must have been mineralised first or was partially oxidised to NO$_3^-$ through the heterotrophic nitrification pathway (Zhang et al., 2015).

To our best knowledge, this is one of the very few studies that document a significant hybrid N$_2$ and N$_2$O production in natural soils without addition of any nucleophiles, i.e., compounds used as the second source of N in codenitrification (Laughlin and Stevens, 2002; Long et al., 2013; Selbie et al., 2015). All these previous studies identified codenitrification as the major N$_2$-producing process, with

contribution of hybrid $N_2$ in the total soil $N_2$ release from 0.32 to 0.95 (Laughlin and Stevens, 2002; Long et al., 2013; Selbie et al., 2015). In our study this contribution is lower, namely 0.18 and 0.05 of the cumulative soil $N_2$ flux, respectively for Min soil and Org soil. No hybrid $N_2O$ was found previously (Laughlin and Stevens, 2002; Selbie et al., 2015), whereas in our study a slight contribution was

detected representing 0.027 and 0.009 of the cumulative $N_2O$ flux for Min soil and Org soil, respectively. Interestingly, we observe higher $f_H$ values for oxic treatments. This may indicate the fungal origin for hybrid $N_2$ and $N_2O$, since it has been shown that fungal denitrification may be activated in presence of oxygen (Spott et al., 2011; Zhou et al., 2001). Similarly, Long et al. (2013) identified fungal codenitrification as the major $N_2$-producing process. In our study, higher $f_H$ values were generally

observed for lower $N_2$ and $N_2O$ fluxes (especially for Org soil, Fig. 2(b), 3(b)). Most probably, towards the end of the incubation, when $N_2$ and $N_2O$ fluxes decrease, also the concentration of intermediate products $NO_2^-$ and NO decrease and the organic substrates may get exhausted. This reinforces the previous observations of enhanced codenitrification for higher ratio between potential nucleophiles and $NO_2^-$ or NO and with decreasing availability of organic substrates (Spott et al., 2011). But we cannot

exclude the possibility that hybrid $N_2$ also originated from other processes, i.e. abiotic codenitrification or annamox (Spott et al., 2011).

A precondition for the proper quantification of various process rates based on the [15]N tracing technique is the homogeneity of [15]N tracer in soil. Recently, a formation of two independent $NO_3^-$ pools in the soil was described for an experimental study (Deppe et al., 2017). One pool contained the

undiluted [15]N tracer solution and thus high [15]N enrichment was mostly the source for $N_2O$. The rest of soil $NO_3^-$ representing the other pool was largely diluted by nitrification input and, therefore, the total soil $NO_3^-$ ($a_{NO3}$) showed lower [15]N enrichment than the [15]N-pool derived $N_2O$ ($a_{P\_N2O}$) (Table 4). This strong discrepancy between pool enrichments could be explained by the large amount of ammonia applied in that experiment and subsequent fast nitrification in aerobic domains of the soil matrix. For

our data, $a_P$ values are not significantly higher than $a_{NO3}$, and for anoxic treatments agree perfectly (Fig. S3.1(a), S3.1(b)), which indicates that the non-homogeneity problem does not apply here. The reason for better homogeneity achieved in our experiments is probably the much higher soil moisture applied,

resulting in more anoxic conditions inhibiting nitrification, and the absence of ammonia amendment. Hence, as we can assume homogenous $^{15}$N distribution, our results on $f_P$ and $f_H$ should be adequate.

## 4.2 N$_2$O isotopic fractionation to quantify N$_2$O reduction

### 4.2.1 Estimating η$_{red}$ and δ$_0$ values

With respect to robust estimation of N$_2$O reduction, a first question arises, to which extent δ$_0$ values and η values were variable or constant during incubations. When assuming constant values of δ$_0$ values during the experiment, calculated η values were highly variable. The large ranges obtained are clearly in strong disagreement with previous knowledge on possible η values (Jinuntuya-Nortman et al., 2008; Lewicka-Szczebak et al., 2014; Ostrom et al., 2007; Well and Flessa, 2009a). In the further interpretation of data we therefore suppose that δ$_0$ values were variable and η values constant. While we cannot rule out that η values varied to some extent, it is not possible to verify that using the current data set.

Another question is whether the assumption of isotopic fractionation pattern of closed systems holds. Logarithmic fits provided best correlations with the measured data, whereas linear correlations that would be indicative for open system dynamics (Decock and Six, 2013) yielded poor fits (data not shown). This indicates that the N$_2$O reduction follows the pattern of a closed system according to Rayleigh distillation equation (Eq. (13)) as suggested previously (Köster et al., 2013; Lewicka-Szczebak et al., 2015; Lewicka-Szczebak et al., 2014).

To which extent are the observed η$_{red}$ and δ$_0$ values in agreement with previous data and how could differences be explained? For Min soil we can compare the η$_{red}$ and δ$_0$ values obtained here to the previous experiment, carried out with the same soil (Exp. 1E, 1F (Lewicka-Szczebak et al., 2014)) but using the acetylene inhibition technique. The actual η$_{red}$$^{15}$N$^{sp}$ values from -8.6 to -6.7 ‰ (Fig. 4(a)) are quite close to that previous result of -6.0 ‰, whereas δ$_0$$^{15}$N$^{sp}$ values from 4.0 to 4.5 ‰ are significantly higher than the previously determined value of -2.7 ‰. While that previous value was within the δ$_0$$^{15}$N$^{sp}$ range of bacterial denitrification (-7.5 to -1.3 ‰, (Toyoda et al., 2005)), the clearly higher actual values indicate that the previous method must have strongly influenced the microbial denitrifying communities, most probably favouring bacterial over fungal denitrification. Much wider ranges of η$_{red}$ values were found for η$_{red}$$^{18}$O (from -22.7 to -9.9 ‰) and η$_{red}$N$^{bulk}$ (from -6.6 to -2.0 ‰, Table 2), which

is also consistent with the previous findings indicating that these values depend on enzymatic and diffusive isotope effects and as result can vary in a quite wide range (Lewicka-Szczebak et al., 2014). The $\eta_{red}$ determined in Exp1 are similar to the previous results (-18 ‰ for $\eta_{red}$ $^{18}O$ and -7 ‰ for $\eta_{red}$ $^{15}N^{bulk}$ (Lewicka-Szczebak et al., 2014)), whereas in Exp2 the absolute values are much smaller, suggesting a different fractionation pattern there. Most probably this difference is an effect of a different range of $r_{N2O}$ in both experiments (Table 2). In Exp2 we partially deal with extremely low $r_{N2O}$ values, which results in smaller overall isotope effects, as also shown before (Lewicka-Szczebak et al., 2015). But $\delta_0^{15}N^{bulk}$ values are very robust since the actual $\delta_0^{15}N^{bulk}$ (-45 ‰, Table 2) corresponds very well to the one previously determined (-46 ‰) using the acetylene method. Conversely, $\delta_0^{18}O$ is much higher (+36 ‰, Table 2) compared to the value of 19 ‰ obtained previously (Lewicka-Szczebak et al., 2014). This may indicate a significant admixture of fungal denitrification characterised by higher $\delta_0^{18}O$ but similar $\delta_0^{15}N^{bulk}$ values (Lewicka-Szczebak et al., 2016; Rohe et al., 2014).

For Org soil, much higher absolute values of $\eta_{red}$ were found (Table 2) being in contrast to all previous studies (Jinuntuya-Nortman et al., 2008; Lewicka-Szczebak et al., 2015; Well and Flessa, 2009a). Hence, it has to be questioned if this observation is not an experimental artefact. Actually, the Org soil anoxic treatment was the only case where $^{15}N$-pool derived $N_2O$ was dominant (Fig. S3.1(b)), hence the isotopic signatures should not be altered due to different $N_2O$ producing pathways but mostly governed by the $r_{N2O}$. But for Org soil, based on the NA treatment, we observe a constant and very significant increase in the contribution of $N_2O$ from fungal denitrification during the incubation (Fig. 5). It should be clarified by future studies if such a rapid microbial shift is possible. Fungal denitrification adds $N_2O$ characterised by higher $\delta^{15}N^{sp}$ values and presumably also higher $\delta^{18}O$ values (Lewicka-Szczebak et al., 2016; Rohe et al., 2014). As a result the $\eta_{red}$ values determined from correlation slopes are biased because the production of $^{18}O$ and $^{15}N^{\alpha}$ enriched $N_2O$ increased in time parallel to a decrease in $r_{N2O}$. In $^{15}N$ treatments this increase in $N_2O$ added from fungal denitrification cannot be distinguished from bacterial denitrification because both originate from the same $^{15}N$ nitrate pool.

The Org soil data thus demonstrate that a high and variable in time contribution of fungal denitrification complicates the application of the $N_2O$ isotopic fractionation approach for quantification of $N_2O$ reduction. This is because a highly variable contribution implies that changes in the measured

$\delta^{15}N^{sp}$ values can either result from variations in $\delta_0^{15}N^{sp}$ or $r_{N2O}$. Only when the contribution of fungal denitrification is stable, robust $r_{N2O}$ values can be derived from $\delta^{15}N^{sp}$ data. Although the Min soil exhibited a smaller range in $f_F$, the contribution of fungal denitrification was apparently also not constant. Simultaneous application of the other isotopic signatures, i.e., $\delta^{15}N^{bulk}$ and/or $\delta^{18}O$, as discussed in further Sect. 4.2.3, may help solving this problem.

### 4.2.2 Calibration and validation of $r_{N2O}$ quantification

The successful calibration shows that $\delta_0^{15}N^{sp}$ and $\eta_{red}$ values were stable enough within Min soil incubation experiments for calculating $r_{N2O}$ using the isotope fractionation approach.

The results of the calibration were very similar if we treated the oxic and anoxic conditions separately and if we used a mean $\eta_{red}$ and $\delta_0^{15}N^{sp}$ value of the oxic and anoxic phase of Exp.1 to all the results (Fig. 6). This indicates that the fractionation factors determined experimentally under anoxic conditions may be applied for isotopic modelling also for oxic conditions, e.g., for parallel field studies in regard to denitrification processes. But importantly, our experiments were performed under high soil moisture and the majority of cumulative $N_2O$ flux also in oxic treatments originated from denitrification (Sect. 3.3), which explains the similar $\delta_0^{15}N^{sp}$ values obtained for oxic and anoxic conditions. For lower soil moisture, differences in $\delta_0^{15}N^{sp}$ values should be expected due to the possible significant admixture of nitrification processes under oxic conditions.

The results of validation show very different agreement between measured and calculated $r_{N2O}$ values depending on the experimental approach used for determination of $\eta_{red}$ and $\delta_0^{15}N^{sp}$ values (Fig.7). When the experiments performed in this study were used (Val2) the agreement was quite good. These experiments are characterised by simultaneous $N_2O$ production and reduction and a longer duration of the experiment of 5 to 9 days. However, when we used values found in a previous experiment using the acetylene inhibition technique (Val1), the agreement is much worse. Estimation of $\eta_{red}$ and $\delta_0^{15}N^{sp}$ using the acetylene inhibition technique included several experimental limitations that might have affected results. Namely, this approach was based on separate parallel experiments with and without $N_2O$ reduction, acetylene amendment required an anoxic atmosphere and the duration of

incubation had to be shorter than 48h. These limitations most probably influence the microbial denitrifying community and do not provide the true $\delta_0{}^{15}N^{sp}$ values.

Whereas finding the true $\delta_0{}^{15}N^{sp}$ values is rather challenging, less problems seem to be related to the $\eta_{red}{}^{15}N^{sp}$ values. For them similar values were found in all the experiments, where He incubations, [5] $^{15}N$ gas flux or acetylene inhibition methods were applied. The determined values were also similar to the mean literature $\eta_{red}{}^{15}N^{sp}$ value of -5 ‰ (Lewicka-Szczebak et al., 2014). Therefore, applying this common literature value for the calculations (Val3) provided also a very good agreement between measured and calculated $r_{N2O}$ values. Hence, this reinforces the previous conclusion that the $\eta_{red}{}^{15}N^{sp}$ value of -5 ‰ can be commonly applied for $r_{N2O}$ calculation (Lewicka-Szczebak et al., 2014), but the [10] major caution should be paid to the proper determination of $\delta_0{}^{15}N^{sp}$ values, which may cause much larger bias of the calculated $r_{N2O}$.

### 4.2.3 Mapping approach to distinguish mixing and fractionation processes

The emitted $N_2O$ is analysed for three isotopocule signatures and the relations between them ($\delta^{15}N^{sp}/\delta^{18}O$, $\delta^{15}N^{sp}/\delta^{15}N^{bulk}$, $\delta^{18}O/\delta^{15}N^{bulk}$) can be informative. Namely, the observed correlation may [15] result from the mixing of two different sources or from characteristic fractionation during $N_2O$ reduction, or from the combination of both processes. If the slopes of the regression lines for these both cases were different, mixing and fractionation processes could be distinguished. Such slopes were often used for interpretations of field data (Opdyke et al., 2009; Ostrom et al., 2010; Park et al., 2011; Toyoda et al., 2011; Wolf et al., 2015) but recently this approach was questioned because of very variable [20] isotopic fractionation noted during reduction for O and N isotopes (Lewicka-Szczebak et al., 2014; Wolf et al., 2015). A recent study showed, that for moderate $r_{N2O}$ (>0.1) the $\delta^{15}N^{sp}/\delta^{18}O$ slopes characteristic for $N_2O$ reduction are quite consistent with previous findings (Lewicka-Szczebak et al., 2015), i.e., vary from ca. 0.2 to ca. 0.4 (Jinuntuya-Nortman et al., 2008; Well and Flessa, 2009a). Hence, in such cases, the reduction slopes may significantly differ from the slopes resulting from mixing of [25] bacterial and fungal denitrification, characterised by higher values of about 0.63 and up to 0.85 (Lewicka-Szczebak et al., 2016).

In theory, the slopes for calculated $\delta_0$ values are not influenced by $N_2O$ reduction and hence should be mostly caused by the variability of mixing processes, whereas the slopes of the measured $\delta$ values reflect both mixing and fractionation due to $N_2O$ reduction. For Min soil, there is no correlation between calculated values of $\delta_0{}^{15}N^{sp}$ and $\delta_0{}^{18}O$ (Table 3), which indicates that the correlation observed for measured $\delta$ values was a result of fractionation processes during $N_2O$ reduction. In contrast, for Org soil all the correlations for calculated $\delta_0$ values are still very strong and show similar slopes as the correlations for measured $\delta$ values (Table 3). This indicates a very significant impact of the mixing of various $N_2O$ producing pathways.

The $\delta^{15}N^{sp}/\delta^{18}O$ slopes for Org soil are generally higher (from 0.65 to 0.76) than for Min soil (from 0.30 to 0.64) (Table 3). This supports the hypothesis from the previous Sect. 4.2.1 about a higher contribution of fungal $N_2O$ in Org soil. But we can also notice that the slopes in Exp1 are lower than in Exp2. Most probably less stable microbial activity is present under the longer incubation in Exp2 (9 days) compared to short phases analysed in Exp1 (3 days). As observed from the calculated $\delta_0$ values (Fig. 5) the estimated contribution of fungal $N_2O$ most probably increases with incubation time. Hence, the higher slopes for Exp2 probably result from the admixture of fungal denitrification and the lower slopes for Exp1 represent more the typical bacterial reduction slopes. The $\delta^{15}N^{sp}/\delta^{18}O$ slopes may thus be helpful in indicating the admixture of various $N_2O$ sources.

Interestingly, there is no correlation between isotopic values in oxic Exp2 for Min soil. A single process or the combination of several processes that cause large variations in $\delta^{15}N^{sp}$ but not in $\delta^{18}O$ seems to be present there. This might be due to admixture of $N_2O$ from different microbial pathways and possibly also due to O-exchange with water. In this treatment we also observe the lowest $N_2O$ fluxes and also the lowest $f_{P\_N2O}$ values, which suggests the largest input from nitrification. The $\delta^{15}N^{sp}$ values for hydroxylamine oxidation during nitrification are much larger (ca. 33 ‰) than for bacterial denitrification or nitrifier denitrification (ca. -5 ‰) (Sutka et al., 2006), whereas $\delta^{18}O$ may be in the same range for both processes (Snider et al., 2013; Snider et al., 2011). This could be an explanation for the missing correlation between $\delta^{15}N^{sp}$ and $\delta^{18}O$ (Table 3).

The graphical interpretations including $\delta^{15}N^{bulk}$ values are more difficult since the isotopic signature of the N precursor must be known, but can be also informative and were often used (Kato et

al., 2013; Snider et al., 2015; Toyoda et al., 2011; Toyoda et al., 2015; Wolf et al., 2015; Zou et al., 2014). The slopes between $\delta^{18}O$ and $\delta^{15}N^{bulk}$ observed in our study range mostly from 1.94 to 3.25 (Table 3), which corresponds quite well to the previously reported results from $N_2O$ reduction experiments where values in the range from 1.9 to 2.6 were reported (Jinuntuya-Nortman et al., 2008;

Well and Flessa, 2009a)). Only for Org soil in anoxic conditions (in both Exp1 and 2) this slope is largely lower and ranges from 0.61 to 0.84. These values are more similar to $\delta^{18}O/\delta^{15}N^{bulk}$ slopes for the calculated $\delta_0$ values (0.56 for Min soil and 1.04 for Org soil (Table 3)) and are significantly lower than typical reduction slopes. Thus, most probably, they are rather due to the mixing of various $N_2O$ sources. However, the calculated $\delta_0$ values cannot be explained with mixing of bacterial and fungal

denitrification only (Fig. S4.3(b)).

      For the relation of $\delta^{15}N^{sp}/\delta^{15}N^{bulk}$ (Fig. S4.2) the reduction and mixing slopes cannot be separated so clearly. The calculated $\delta_0$ values are not all situated between the mixing endmember of bacterial and fungal denitrification. This observation is similar as for $\delta^{18}O/\delta^{15}N^{bulk}$ and is due to some data points showing very low $\delta_0{}^{15}N^{bulk}{}_{(N2O/NO3\text{-})}$ values down to ca. -70 ‰. This value exceeds the

known range of the $^{15}N$ fractionation factors due to the $NO_3{}^-/N_2O$ steps of denitrification, i.e., based on pure culture studies, from -37 to -10 ‰ for bacterial and from -46 to -31 ‰ for fungal denitrification (Toyoda et al., 2015) (as displayed on graphs in Fig. S4) and, based on controlled soil studies, from -55 to -24 ‰ (Lewicka-Szczebak et al., 2014; Well and Flessa, 2009b). This additional $N_2O$ input may originate from nitrifier denitrification, as already suggested based on the $^{15}N$ treatments results (Sect.

3.3). Frame and Casciotti (2010) determined fractionation factors for nitrifier denitrification: $\varepsilon^{15}N^{bulk}{}_{NH4/N2O}$ = 56.9 ‰, $\varepsilon^{18}O_{N2O/O2}$ = -8.4 ‰ and $\varepsilon^{15}N^{SP}$ = -10.7 ‰. When recalculated for values presented in our study, $\delta_0{}^{18}O_{N2O/H2O}$ will range from 22 to 25 ‰ (taking the variations in $\delta^{18}O_{H2O}$ into account). Unfortunately, the $\delta_0{}^{15}N^{bulk}$ value for this process could not be assessed in our study, since the $\delta^{15}N_{NH4}$ was not measured. In case the $\delta^{15}N_{NH4}$ is lower than 0 ‰, the very low $\delta_0{}^{15}N^{bulk}{}_{(N2O/NO3\text{-})}$ values

may be well explained with nitrifier denitrification.

      Although the interpretation of the relations between particular isotopic signatures is not completely clear yet, it seems to have a potential to differentiate between mixing and fractionation processes. Note that by using the literature ranges of isotopic end-member values, they must be

recalculated according to respective substrate isotopic signatures for the particular study, hence $\delta^{15}N_{NH4}$, $\delta^{15}N_{NO3}$ and $\delta^{18}O_{H2O}$ should be known. Only the $\delta_0{}^{15}N^{sp}$ can be directly adopted. Progress in interpretations could be made if all three isotopic signatures would be evaluated jointly in a modelling approach. In order to produce robust results, precise information on $\delta_0$ values for all possible $N_2O$ source processes must be available for the particular soil. Unfortunately, the complete modelling is not possible for the data presented here as information on the $NH_4{}^+$ isotopic signature and the $\delta_0{}^{15}N^{bulk}$ value for possible nitrification processes is lacking.

The mapping approach had been used before based on $\delta^{15}N^{sp}$ and $\delta^{15}N^{bulk}$ to estimate the fraction of bacterial $N_2O$ (Zou et al, 2014). Because $N_2$ fluxes were not measured in that study, scenarios with different assumptions for $N_2O$ reduction were applied to show the possible range of the bacterial fraction. Here, we evaluated the mapping approach for the first time using independent estimates of $N_2O$ reduction. Most informative are the relations between $\delta^{15}N^{sp}$ and $\delta^{18}O$, because $\delta_0{}^{15}N^{bulk}$ was poorly known, whereas the estimation of $\delta_0{}^{18}O$ is quite robust due to the large O-exchange with water and constant fractionation during O-exchange as shown previously (Lewicka-Szczebak et al., 2016). Therefore we proposed here a method based on $\delta^{15}N^{sp}$ and $\delta^{18}O$ values to calculate simultaneously the $N_2O$ residual fraction ($r_{N2O}$) and the contribution of the mixing end-members as described in 2.7.3. From Fig. 8 we can assume that the method works quite well in case of a significant admixture of fungal $N_2O$ and allows quantifying its fraction ($f_F$). For the three treatments where a good agreement between measured and calculated $r_{N2O}$ is observed, we rather deal with a significant contribution of fungal $N_2O$ (Sect. 4.2.1). The $f_F$ values calculated here from the mapping approach are very consistent with the values found based on estimated $\delta_0{}^{15}N^{sp}$ only (Fig. 5), i.e. without considering $\delta^{18}O$ values. In the oxic Min soil treatment we probably deal with significant contribution of $N_2O$ originating from nitrification or nitrifier denitrification, as supposed previously from the [15]N treatment (Sect. 4.1) and from the isotopic relations discussed above. The oxic Min soil treatment thus results in rather poor agreement of the mapping approach results. The combination of these processes seems to be too complex to precisely quantify their contribution in $N_2O$ production based on three isotopocule signatures only.

Importantly, for Org soil where $f_F$ values are very high and variable with time (see also Sect. 4.2.1), the mapping approach was the only method to get any estimation of both $f_F$ and $r_{N2O}$. The other approach, presented in Sect. 2.7.2 and successfully applied for Min soil, failed for Org soil due to the inability to assess a stable $\delta_0{}^{15}N^{sp}$. Hence, for the case of varying contribution of fungal $N_2O$, the mapping approach presented here may be the only way of assessing the range of possible $f_F$ and $r_{N2O}$ values. However, the precision of the results obtained from the mapping approach is a complex issue depending on size of endmembers areas and variability of η values. We did not aim to determine the resulting uncertainty in the present paper. The following paper will address the precision problem in detail (Buchen et al., in preparation).

## Conclusions

We have shown that the $N_2O$ isotopic fractionation approach based on $\delta^{15}N^{sp}$ values is suitable to identify and quantify $N_2O$ reduction under particular conditions, most importantly, quite stable $N_2O$ production pathways. It has been confirmed that the range of $\eta_{red}{}^{15}N^{sp}$ values defined in previous studies is well applicable for the calculations. The calculated $N_2O$ residual fraction is much more sensitive to the range of possible $\delta_0{}^{15}N^{sp}$ values rather than $\eta_{red}{}^{15}N^{sp}$ values. Therefore, $\delta_0{}^{15}N^{sp}$ values must be determined with large caution. The method can be used in field studies, but to obtain robust results, in situ measurement of isotopocule fluxes should be complemented by laboratory determinations of $\delta_0{}^{15}N^{sp}$ values. For this aim, the He incubation technique or the $^{15}N$ gas flux method can be applied as reference methods, but not the acetylene inhibition method, since it most probably affects the microbial community, which results in biased $\delta_0{}^{15}N^{sp}$ values. Anoxic incubations may be applied and the determined $\delta_0{}^{15}N^{sp}$ values are representative for $N_2O$ originating from denitrification, also for oxic conditions, which means, also in field studies.

The attainable precision of the method, determined as mean absolute difference between the measured and the calculated $N_2O$ residual fraction ($r_{N2O}$), is about ±0.10, but for individual measurements this absolute difference varied widely from 0.00 up to 0.39. The relative error of $N_2$ flux quantification depends strongly on the $r_{N2O}$ of a particular sample and varied in a very wide range from 0.01 up to 2.41 for Exp1 and from 0.00 up to 0.93 for Exp2, with a mean relative difference between

measured and calculated $N_2$ flux of 0.46 and 0.13, respectively. The highest relative errors in the calculated $N_2$ flux (>1) occur for the very low fluxes only ($r_{N2O} > 0.9$).

However, for soils of more complex N dynamics, as shown for the Org soil in this study, the determination of $N_2O$ reduction is more uncertain. The method successfully used for Min soil was not applicable due to failed determination of proper $\delta_0{}^{15}N^{sp}$ values, which were significantly changing with incubation progress. Here we suggest an alternative method based on the relation between $\delta^{15}N^{sp}$ and $\delta^{18}O$ values ('mapping approach'). This allows for the estimation of both the fraction of fungal $N_2O$ and the plausible range of residual $N_2O$.

**Acknowledgements**

This study was supported by German Research Foundation (DFG: We/1904-4, LE 3367/1-1). Many thanks are due to Martina Heuer for help in $N_2O$ isotopic analyses; Bertram Gusovius for assistance by helium incubations; Stefan Burkart for assistance by microcosm incubations; Kerstin Gilke for help in chromatographic analyses, Roland Fuß for advice in statistical evaluation, and Caroline Buchen for supplying soil for laboratory incubations.

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

**Tables:**

**Table 1: Rates of N transformation processes as calculated from $^{15}$N-pool dilution for Exp2 $^{15}$N treatment. Source measured data used for the calculation are provided in the supplement (Table S1).**

| treatment | N-transformations: calculated rates [mg N per kg dry soil per 24h] | | | | | |
|---|---|---|---|---|---|---|
| | nitrification | unlabelled $N_2O$ flux $f_{N\_N2O} \times [N_2O]$ | DNRA | mineralisation | total N-gas flux $[N_2+N_2O]$ | immobilisation |
| Min Soil | | | | | | |
| oxic | 0.30 | 0.01 | b.d. | 0.31 | 0.02 | 2.18 |
| anoxic | 0.05 | 0.04 | 0.02 | 0.15 | 1.67 | 2.51 |
| Org Soil | | | | | | |
| oxic | 1.93 | 0.07 | b.d. | 1.99 | 0.34 | 6.29 |
| anoxic | 0.06 | 0.13 | 0.10 | 1.25 | 10.42 | 9.53 |

5    b.d. - $^{15}$N below detection limit

**Table 2: Fractionation factors of N₂O reduction ($\eta_{red}$) and isotopic signatures of initial unreduced N₂O ($\delta_0$) determined from the regression function $\delta = \eta_{red} \times \ln(r_{N2O}) + \delta_0$ (Eq. (14)). Statistical significance given for $\alpha=0.05$ with \*$p<0.05$,\*\* $p<0.01$, \*\*\*$p<0.001$ from Pearson correlation coefficients.**

| | $\delta^{18}O(N_2O/H_2O)$ | | $\delta^{15}N^{bulk}(N_2O/NO_3^-)$ | | $\delta^{15}N^{sp}$ | | $r_{N2O}$ range |
|---|---|---|---|---|---|---|---|
| | $\eta_{red}$ | $\delta_0$ | $\eta_{red}$ | $\delta_0$ | $\eta_{red}$ | $\delta_0$ | |
| Min soil, Exp1 | | | | | | | |
| anoxic | -15.5 ** | +35.7 ** | -6.6 ** | -48.7 ** | -8.6 *** | +4.4 *** | 0.19 - 0.75 |
| oxic | -22.7 *** | +37.0 *** | -5.7 *** | -42.0 *** | -6.8 *** | +4.5 *** | 0.27 - 1.00 |
| Min soil, Exp2 | | | | | | | |
| anoxic | -9.9 *** | +35.5 *** | -2.0 *** | -45.2 *** | -6.7 *** | +4.0 *** | 0.01 - 0.59 |
| oxic | n.a. | n.a. | n.a. | n.a. | n.a. | n.a. | 0.04 - 0.71 |
| Org soil, Exp1 | | | | | | | |
| anoxic | n.a. | n.a. | n.a. | n.a. | n.a. | n.a. | 0.30 - 0.84 |
| oxic | n.a. | n.a. | n.a. | n.a. | n.a. | n.a. | 0.05 - 0.56 |
| Org soil, Exp2 | | | | | | | |
| anoxic | -38.4 *** | +20.6 *** | -32.9 *** | -60.9 *** | -30.8 *** | -3.4 *** | 0.09 - 0.82 |
| oxic | -25.4 *** | +24.6 *** | -6.8 * | -47.1 * | -20.8 *** | -3.3 *** | 0.10 - 0.88 |

5    n.a. - not applicable - no statistically significant correlation

**Table 3: Relations between isotopic signatures of emitted N$_2$O: $\delta^{15}N^{sp}/\delta^{18}O$; $\delta^{15}N^{sp}/\delta^{15}N^{bulk}$; $\delta^{18}O/\delta^{15}N^{bulk}$ and mean $r_{N2O}$ of the corresponding data-sets. The slopes for linear fit are given. Statistical significance given for $\alpha=0.05$ with *$p<0.05$, **$p<0.01$, ***$p<0.001$ from Pearson correlation coefficients. The graphical presentation of the correlations is shown in the supplement (Fig. S4)**

| | $\delta^{15}N^{sp}/\delta^{18}O$ | $\delta^{15}N^{sp}/\delta^{15}N^{bulk}$ | $\delta^{18}O/\delta^{15}N^{bulk}$ | $r_{N2O}$ mean |
|---|---|---|---|---|
| | slope | slope | slope | |
| **Min soil, Exp1** | | | | |
| Anoxic | 0.47 *** | 1.01 *** | 2.21 *** | 0.46 |
| oxic | 0.30 *** | 0.59 *** | 1.94 *** | 0.77 |
| **Min soil, Exp2** | | | | |
| anoxic | 0.64 *** | 2.16 *** | 3.25 *** | 0.14 |
| oxic | n.a. | n.a. | n.a. | 0.39 |
| **Org soil, Exp1** | | | | |
| anoxic | 0.65 *** | 0.55 *** | 0.84 *** | 0.59 |
| oxic | n.a. | n.a. | n.a. | 0.34 |
| **Org soil, Exp2** | | | | |
| anoxic | 0.76 *** | 0.82 *** | 0.61 *** | 0.48 |
| oxic | 0.73 *** | 2.07 *** | 3.07 *** | 0.44 |
| **Min soil, all data** | | | | |
| calculated $\delta_0$ | n.a. | n.a. | 0.56 ** | |
| **Org soil, all data** | | | | |
| calculated $\delta_0$ | 0.68 *** | 0.74 *** | 1.04 *** | |

n.a. - not applicable - no statistically significant correlation

**Table 4: Results from a laboratory incubation experiment to distinguish between $N_2O$ emitted from nitrificaion and denitrification in a sandy loam soil (Deppe et al., 2017) in comparison with this study results (Min and Org soil). Results of Deppe et al. (2017) show large differences between average $^{15}N$ enrichment of $NO_3^-$ in the bulk soil as analysed in extracted $NO_3^-$ and $^{15}N$ enrichment of $NO_3^-$ in denitrifying microsites producing $N_2O$ as calculated from the non-equilibrium approach after Spott et al. (2006) and Bergsma et al. (2001).**

| | Deppe et al., 2017 | Min soil, oxic | Org soil, oxic | Min soil, anoxic | Org soil, anoxic |
|---|---|---|---|---|---|
| $a_{NO3}$ of added fertilizer | 12.5 | 51.1 | 43.2 | 51.1 | 43.2 |
| $a_{NO3}$ at final sampling | 2.24±0.02 | 49.6±0.1 | 39.9±0.2 | 50.8±0.2 | 43.0±0.2 |
| $a_{P\_N2O}$ at final sampling | 13.0±0.9 | 47.7±0.5 | 37.2±1.0 | 51.2±0.1 | 45.9±0.3 |
| $a_{P\_N2}$ at final sampling | n.d. | 49.3±1.5 | 38.7±1.0 | 49.8±0.4 | 43.3±1.3 |

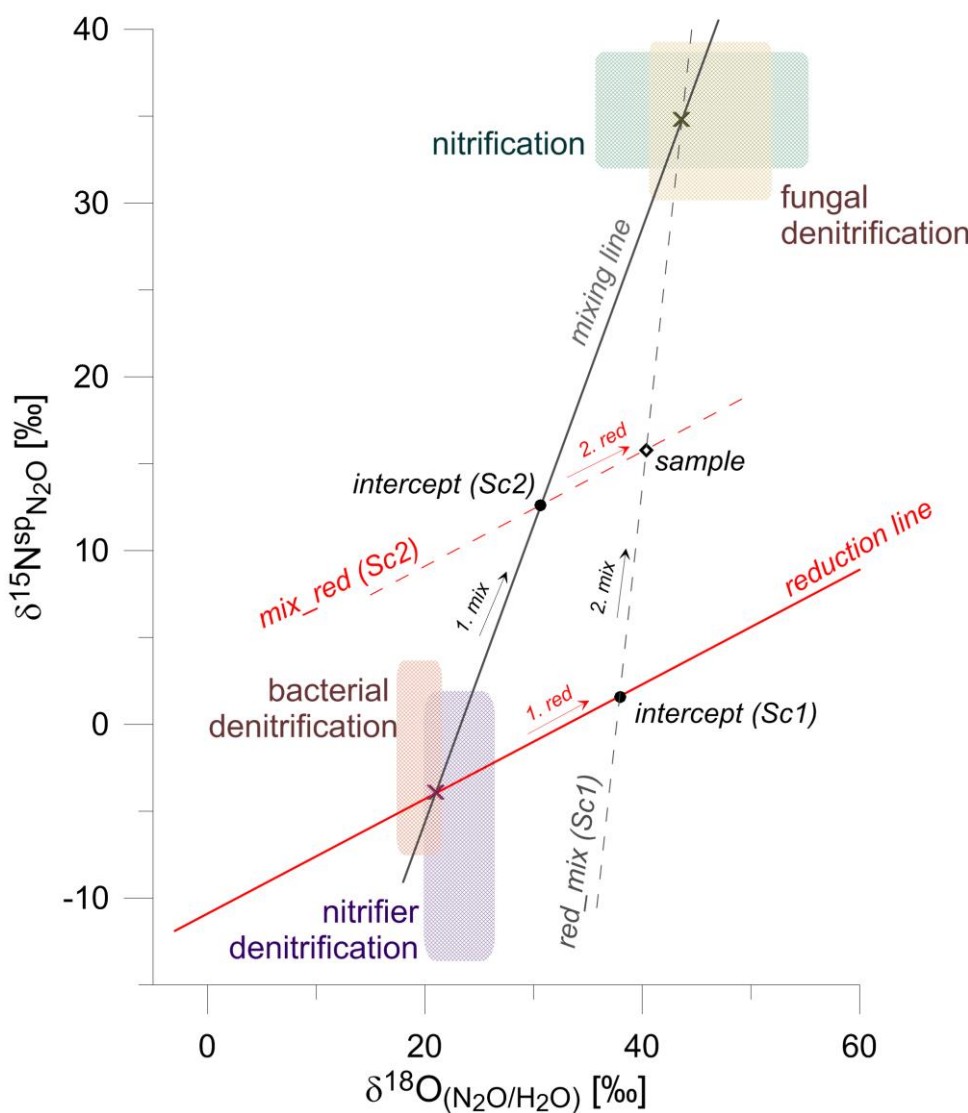

**Figure 1: Scheme of the mapping approach to simultaneously estimate the magnitude of N₂O reduction and the admixture of fungal denitrification (or nitrification).**

**Figures:**

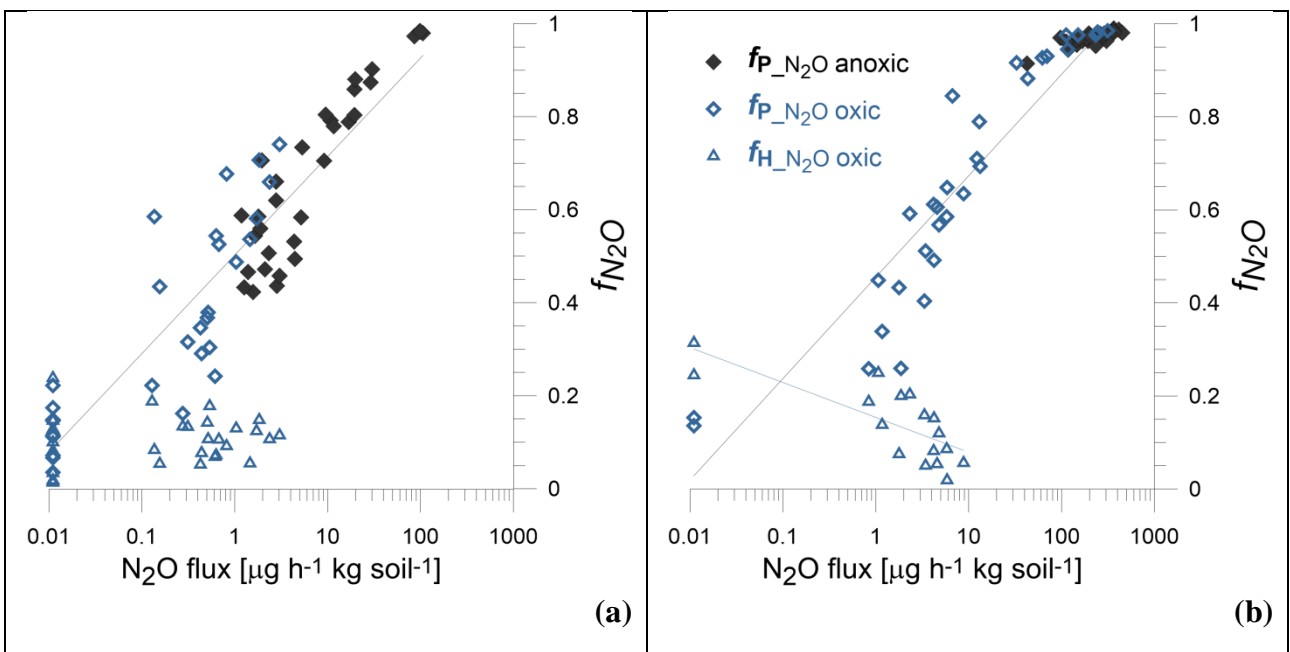

**Figure 2: Contribution of $^{15}$N-pool derived N$_2$O in the total N$_2$O flux ($f_{P\_N2O}$ - diamonds) and the fraction of hybrid N$_2$O within the $^{15}$N-pool derived N$_2$O ($f_{H\_N2O}$ - triangles) in relation to the total N$_2$O flux for Min (a) and Org (b) soil in oxic (blue data points) and anoxic (black filled data points) conditions. No hybrid N$_2$O was detectable under anoxic conditions. Logarithmic correlation is shown where statistically significant ($f_P$ Min soil: $R^2$=0.80, $p$<0.001; $f_P$ Org soil: $R^2$=0.88, $p$<0.001; $f_H$ Org soil: $R^2$=0.59; $p$=0.013). Fluxes lower than 0.01 (detection limit) are shown jointly as <0.01.**

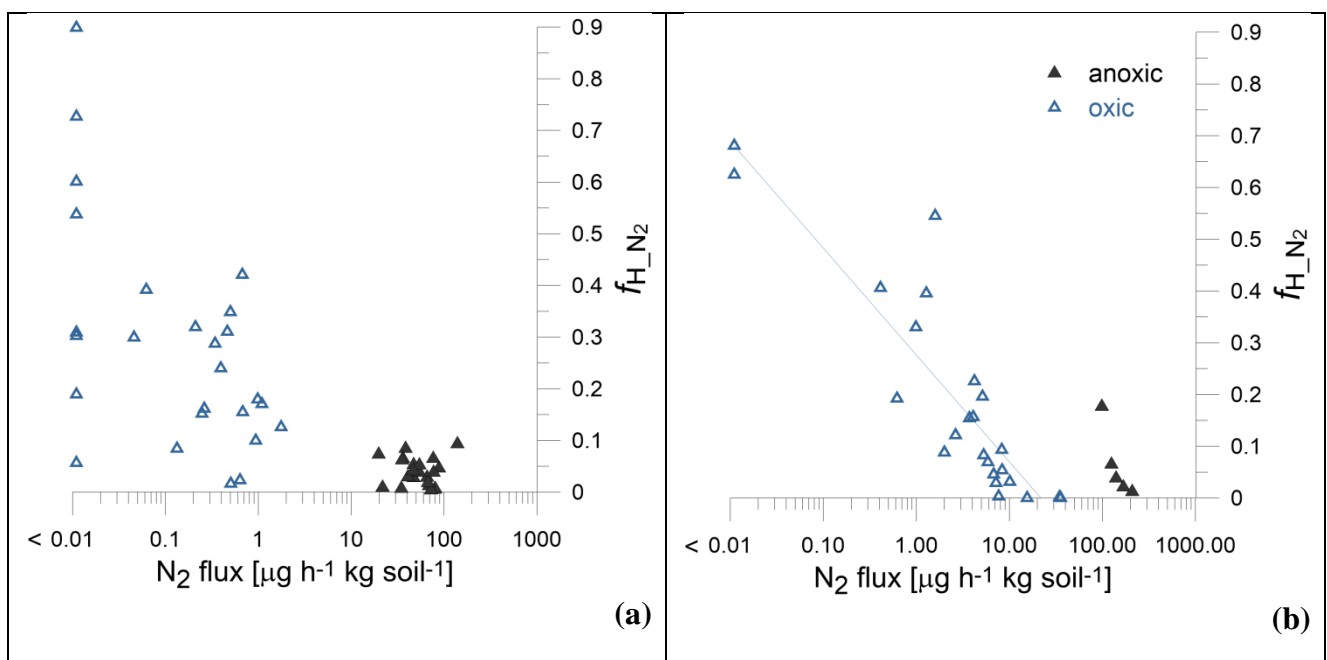

**Figure 3: Contribution of hybrid N$_2$ in the total $^{15}$N-pool derived N$_2$ in relation to the N$_2$ flux for Min (a) and Org (b) soil under oxic (blue triangles) and anoxic (black triangles) conditions. Logarithmic correlation is shown where statistically significant (f$_H$ Org soil oxic: $R^2$=0.79; p<0.001). Fluxes lower than 0.01 (detection limit) are shown jointly as <0.01.**

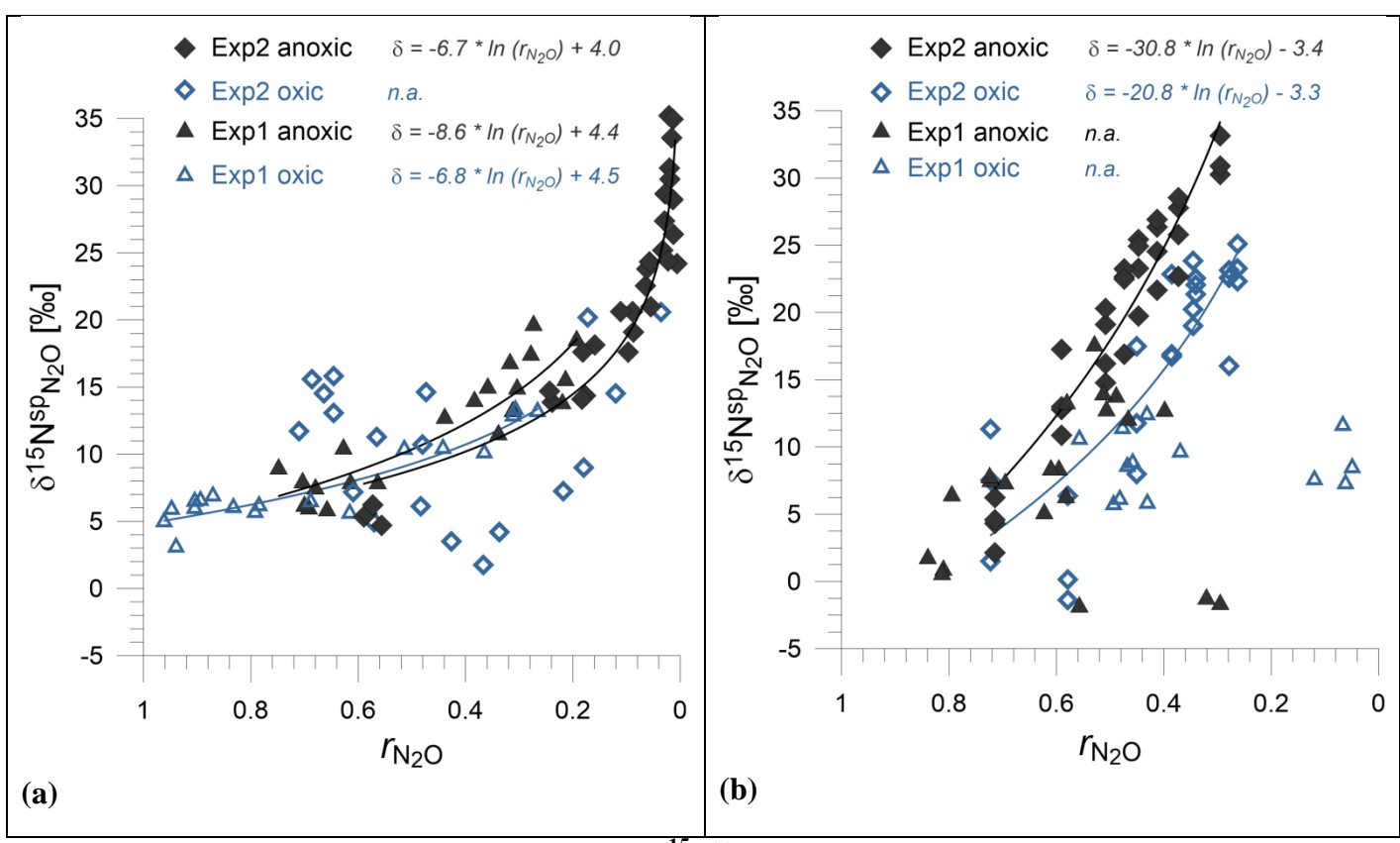

**Figure 4: Examples for the relation between $\delta^{15}N^{sp}$ and $r_{N2O}$: Min soil (a) and Org soil (b). The equation for ln correlations are given where significant, *n.a.* where not significant.**

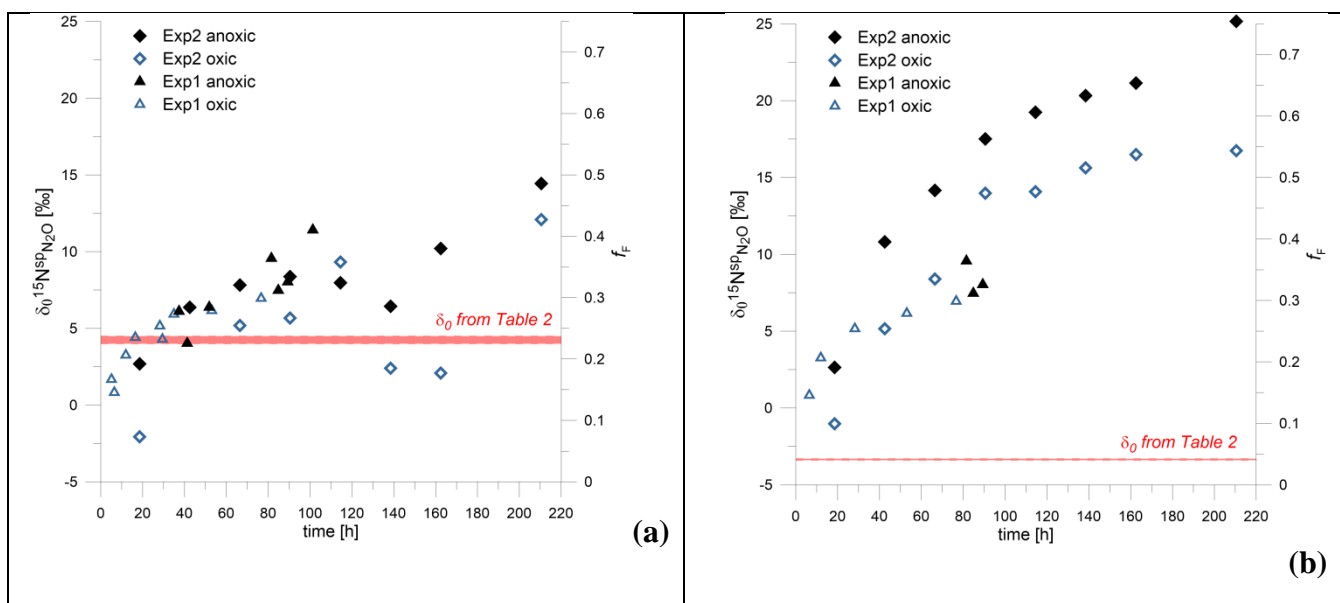

**Figure 5:** Calculated $\delta_0{}^{15}N^{sp}$ values for individual samples (assuming common stable $\eta_{red}{}^{15}N^{sp}$ value of -5 ‰) with the respective fraction of fungal $N_2O$ ($f_F$) (calculated with endmembers $\delta_0{}^{15}N^{sp}$ values: -5 ‰ for bacterial and 35 ‰ for fungal denitrification). The individual $\delta_0{}^{15}N^{sp}$ values are compared with the general $\delta_0{}^{15}N^{sp}$ value calculated from the overall correlation between $\delta^{15}N^{sp}$ and $r_{N2O}$ (Table 2). Min soil (a) and Org soil (b).

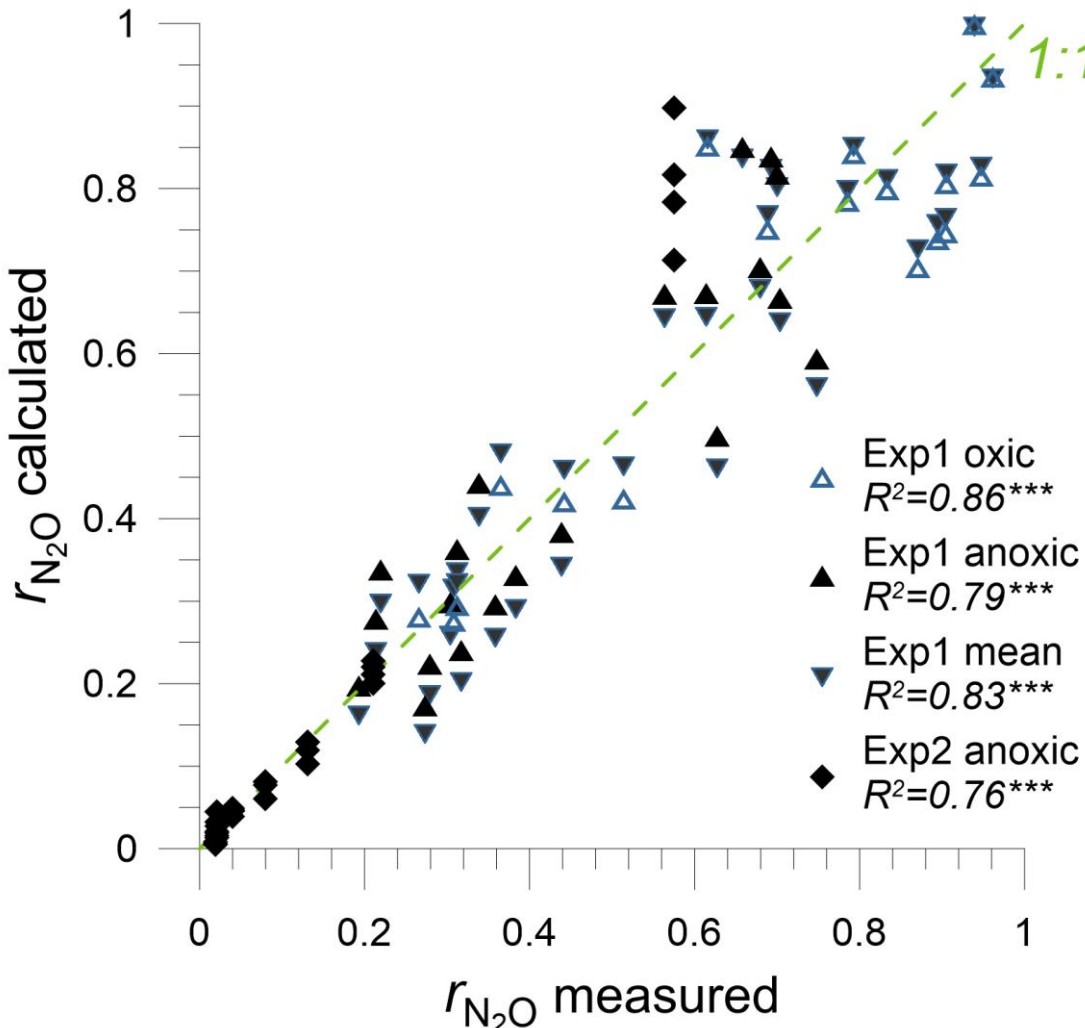

**Figure 6: Calibration of the N$_2$O isotopic fractionation approach using Min soil data.** $r_{N2O}$ **calculated based on Eq. (17) and measured with independent methods are compared. For Exp1 the values calculated based separately either on an oxic (blue triangles) or an anoxic treatment (filled black triangles) or based on the mean values (reversed blue triangles) are shown. For Exp2 only anoxic treatment samples are shown, since for oxic treatment the relevant reference data is missing (see discussion in 3.4.1)**

**Goodness of fit to the 1:1 line is expressed as R$^2$ and the statistical significance is determined for $\alpha$=0.05 with \*p<0.05,\*\* p<0.01, \*\*\*p<0.001 from Pearson correlation coefficients.**

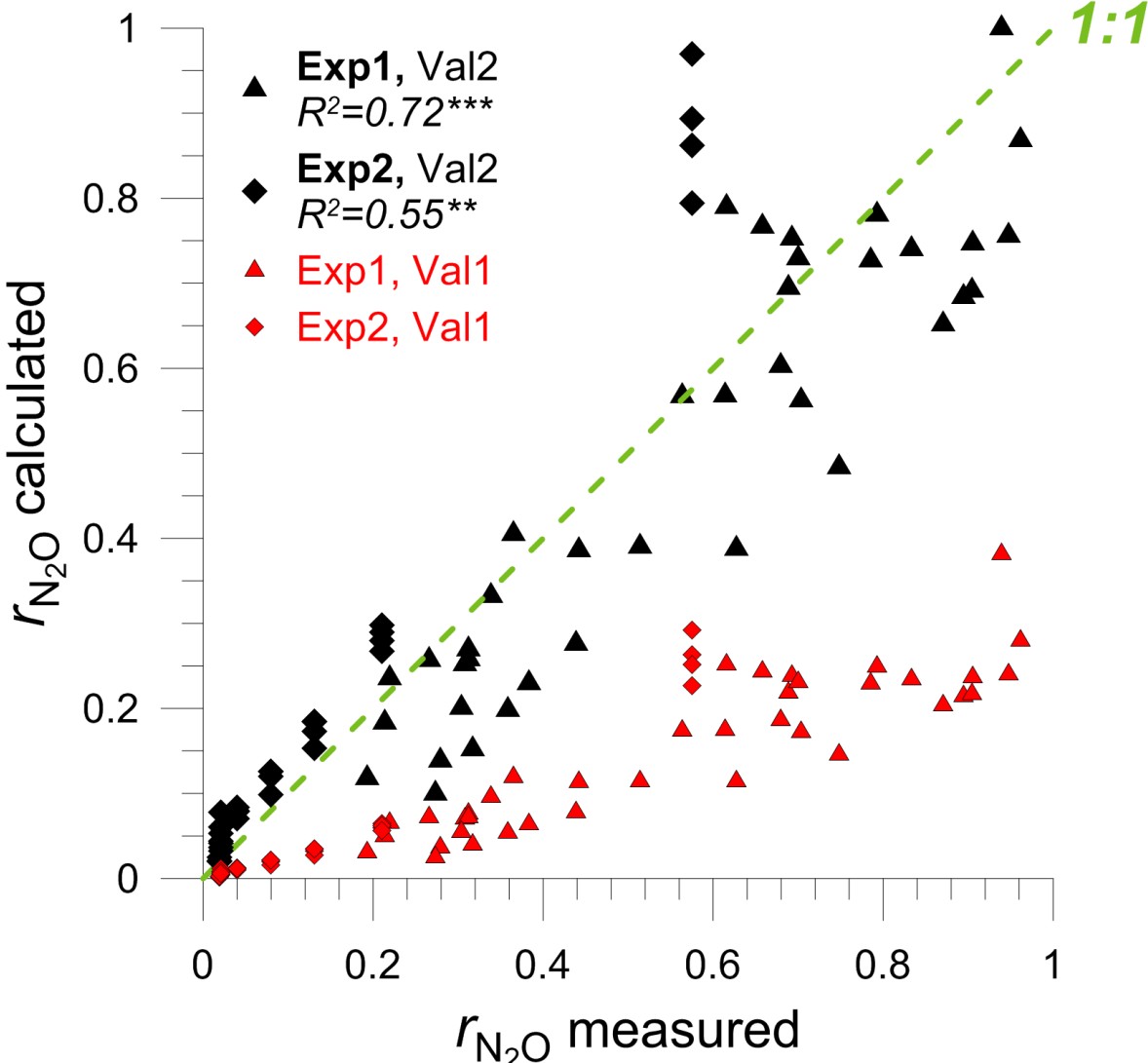

**Figure 7: Validation of the N$_2$O isotopic fractionation approach using Min soil data.** $r_{N2O}$ **calculated based on Eq. (17) and measured with independent methods are compared. For Exp1 (triangles) and Exp2 (diamonds) the values calculated based on previous static experiment (Val1 - red points) and on this study (Val2 - black points) are shown.**
**Goodness of fit to the 1:1 line is expressed as R$^2$ and the statistical significance is determined for $\alpha$=0.05 with \*$p$<0.05,\*\* $p$<0.01, \*\*\*$p$<0.001 from Pearson correlation coefficients.**

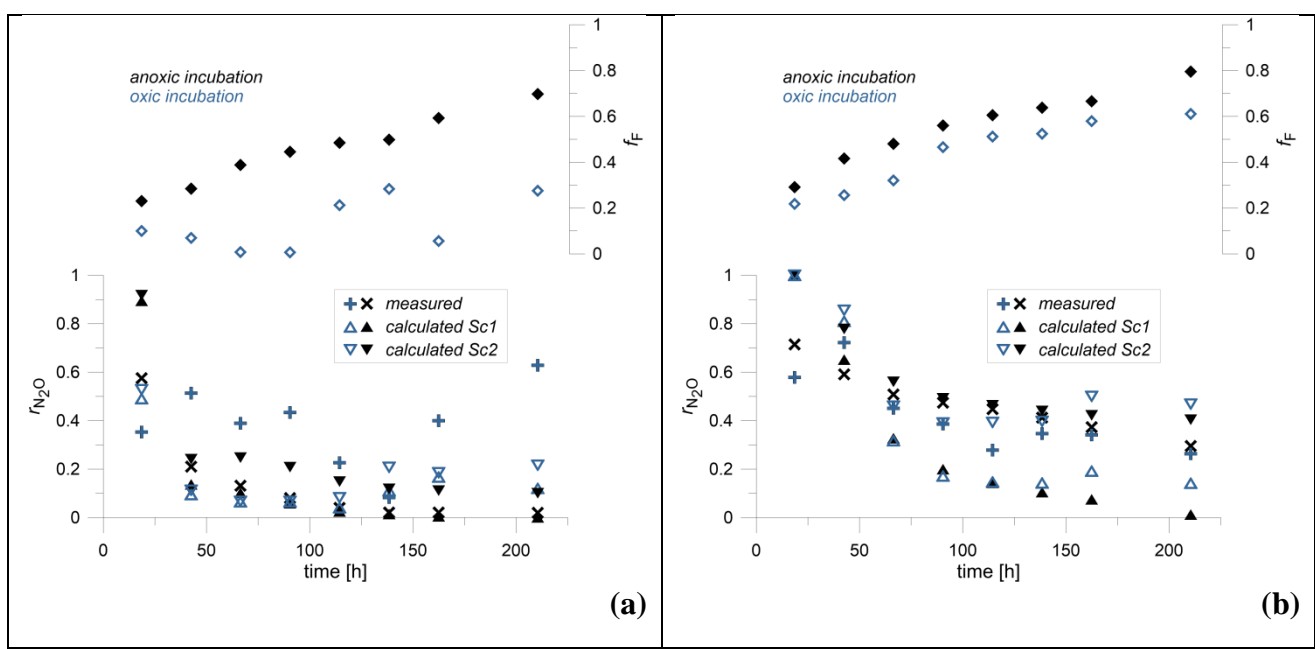

**Figure 8: The calculated contribution of N$_2$O originating from fungal denitrification or nitrification ($f_F$, upper graph, diamonds) and the calculated residual N$_2$O fraction ($r_{N2O}$) with two scenarios (triangles) compared to the measured values (crosses). Filled black symbols represent anoxic incubation and open blue symbols - oxic incubation. Min soil (a) and Org soil (b).**