# Peer review of "Quantifying $N_2O$ reduction to $N_2$ based on $N_2O$ isotopocules – validation with independent methods (helium incubation and $^{15}N$ gas flux method)"

_Biogeosciences, 2016_

## Referee Comment (RC1) · Anonymous Referee #1 · 20 Sep 2016

**1. General comments**

This paper examines the validity of isotopic analyses to quantify the degree of N2O reduction to N2 occurring in soils. Molecular nitrogen is the final product in denitrification, one of the major nitrogen metabolism processes in soils, but the rate of N2 production under natural condition is usually difficult to determine due to the presence of atmospheric N2. Isotopic fractionation of N2O, a precursor of N2 in denitrification, has been often used as an alternative to estimate the N2 production rate. However, isotopic signature of N2O is determined not only by N2O reduction but also N2O production by denitrification and other microbial or abiotic processes, so there remains significant uncertainty with the isotopic method for N2 production estimates. Based on independent

laboratory simulation experiments, the authors showed that isotopic signature of N2O before reduction is the most critical parameter and it can be experimentally determined under steady conditions and that combined analysis of two isotopic values is effective under non-steady conditions.

Although the authors' major finding is somewhat predictable and the "mapping approach" is not really new compared to previous studies, this work is the first one to confirm the validity of isotopic fractionation method experimentally and it contains some useful information on experimental approaches and on various N2O production processes occurring in soils. Therefore, I consider that this paper is suitable for publication in BG after minor revision, and I expect further researches are stimulated by this paper to apply the isotopic method to more complicated soil systems.

2. Specific comments

P9, L13 and eq. 5: It is not clear whether the N2 originating from non-labelled pools are considered similarly in the case of N2 because evidence of such N2 is described in section 3.2.2.

P9, eq. 6: In P8, the authors describe that three separated gas species (N2, N2+N2O, and N2O) were measured. Then why is fp_N2O not used in this equation? Or did they confirm mass balance like fp_N2+N2O = fp_N2 + fp_N2O?

P11, L4: "average 15N abundance in nitrate" What was averaged? Initial and final values?

P18, section 3.2.2: Although the calculation procedure for r_N2O is explained in detail in section 2.5.2, results of r_N2O from the 15N treatment are not shown. Does this mean the 15N gas flux method failed to give r_N2O value?

P23, L6: "r_N2O values are always higher for Sc2" This is consistent with Figure 8, but I found the opposite statement in P16, L19. Please check the text and the figure.

P23, L23: "for both soils in the anoxic treatment the cumulative non-labelled N2O flux

is higher than the initial NH4+ pool plus the NH4+ possibly added" I could not follow. When I compare the 8th, 12nd, and 13th columns in Table S1, this is the case only for Min Soil with anoxic, 15N treatment.

P24, L6: "it represents, respectively, 2 and 3 % of the nitrification rate (Table 1)" I cannot understand how I find this in Table 1. What does "respectively" means?

P24, L8: "observed increase in NO3-" In Table S1, C_NO3t is always lower than C_NO30, so there seems to be no "increase in NO3-".

P27, L8: "15N-pool derived N2O characterized by higher d15Nsp values" In section 2.5, the authors did not mention that they measured d15Nsp of N2O in 15N gas flux method. Did they measure it?

P32, L27: It seems that 15N gas flux method is useful to detect the processes such as producing hybrid N2O or N2, but I'm not convinced that it is really necessary to determine r_N2O (see above comment on section 3.2.2).

3. Technical corrections

P9, L3: "..., of which the fp was derived" "from which" would be better.

P10, equations 11 and 12: To be exact, eq. 12 is not automatically derived from eq 11. I suggest to define f_L together with f_H.

P11, L8, L9, L13: Be consistent in expressing "[N2O flux]". It was [N2O]flux in P10, eq 7.

P12, L10: "d15Nbulk(NO3/N2O)": positions of NO3 and N2O should be reversed?

P14, L11: Which do you prefer to use "SP" or "d15Nsp"? Consistent usage is preferable.

P14, L11: "... have been use to" ... have been used to

P21, L26: "no significant correlation with the 1:1 line" Awkward phrase. I suppose the

authors would like to say that slope was smaller than unity although calculated and measured values showed a correlation.

P23, L16: "Fig. 3", L23 "Fig. 1(a)", P24, L5 "Fig. 1" Are these referring to Fig. 2? Also I suggest to rearrange the figures in order of mention in the text.

P27, L20: "Sect. 4.2.3" Sect. 4.3?

P28, L20: "For them . . ." Awkward sentence.

Caption of Figure 6: "Eq. (13)" might be Eq. (17). Also, it should be clearly noted that this graph shows data obtained only with Min Soil.

Caption of Figure 7: Correct the equation number in "Eq. (13)".

---

## Referee Comment (RC2) · Anonymous Referee #3 · 2 Nov 2016

General Comments

The manuscript entitled, "Isotope fractionation of N2O to quantify N2O reduction to N2- validation with Helium incubation and 15N gas flux methods," details a series of experiments to validate a new method for determining the amount of N2O reduction through mapping the isotopic values of N2O. This method was compared to two types of traditional incubation methods. I think that this work is substantial and would add to the field of N2O isotopes.

While I agree with the science, I found the presentation of the material in writing and figures to be poor, and the information barely accessible to the reader. I had a very difficult time understanding parts of the manuscript, and believe it should not be published

as it is. The sentences were often long and awkwardly written. Additionally, many of the terms were not well defined and described early in the manuscript. Then when the results were presented it was difficult to understand what each variable meant, and why they were important. The figures were also cluttered with too much information. I would have liked to seen there be more things distilled down for the reader, rather than showing all the data and every experiment. I wonder if all the experiments or data should be presented in one paper or if some of this information would be best split into multiple manuscripts.

With all that said, I do think there is intellectual merit in the experiments that were preformed and the results they found. I think the "mapping approach" presented would be a more efficient and easy way to determine N2O reduction in the environment rather than using incubation methods that are expensive, time consuming and riddled with caveats. In the manuscript they also defined the clear weak points of this new method, namely knowing the $\delta 0$, which will steer future research to improve the method in this area. With substantial revision this paper will be a great addition to the field.

Specific Comments

P1 L1- I would suggest editing the title to make it catchier and less awkward.

P1 L10- Rephrase, "the main unknown magnitude"

P1 L11- Define in the abstract what the residual fraction is.

P2 Introduction- Add more description on the importance of being able to determine N2O reduction. Also, give a better background on all the important terms to be later used in the manuscript and why they are important. A figure or table might be helpful for showing previous work and how the terms fit into the overall picture.

P2 L4-6, P3 L1-3, P3 L26-28- Long and awkward sentences, consider rewriting.

P4 L19- The heading title has "experiment 1 and Exp 1" I would only write it once in the title.

P5 L2- Missing "a" in "application of a N2-free atmosphere"

P5 L15- Fix section heading, as above.

P5 L21- Why is the nitrate treatment different in Exp2 than Exp1?

P6 L8- Define NA.

Supplemental Figures S1 and S2- I suggest removing some of the variables from the figures and putting a simplified figure in the main text. I was also confused with the labeling and order of the figure S2, such that it went from 2.1 a) to 2.2 a) and then back to 2.1 b), could you combine panels onto 1 page and make them a b c and d?

P14 L24- In the "mapping approach" how much will the answer change if you use different end member values? The boxes for possible values are large and suspect it could be large.

P15 L2-7: What is the value are you referring to in this paragraph?

P16 L1: I was surprised that the reduction isotope ratios were the same for oxic and anoxic incubations. Why is that?

P19 L3: What is N immobilization?

P19 L17: What is hybrid N2O? Could you define it earlier in the manuscript?

P20 L4: What correlations?

P21-22: What are the differences between Val 1 and Val 2? Can you state them more clearly before presenting the results?

P22 L25: Could the data in Table 3 be put into a simplified graph in the main text? That might be helpful for the reader.

P25 L10-15: I'd suggest putting the historical data in a table with the current findings.

P27 L22: The title "Calibration and Validation" is vague, calibration and validation of

what?

Table 1: I suggest putting the full names of the variables in the table header row.

Figure 2 and 3: Why is the x-axis on the right hand side?

Figure 4, 6 and 8: Symbols are similar and hard to distinguish in the figure.

Figure 5: There is no legend.

---

## Author Comment (AC1) · 4 Dec 2016

***Review response on*** "Isotopic fractionation of N2O to quantify N2O reduction to N2 – validation with Helium incubation and 15N gas flux methods" ***by*** **Dominika Lewicka-Szczebak et al.**

**Anonymous Referee #1**

Referee comments in red,
authors response in black.

Thank you very much for reviewing our manuscript and for your valuable suggestions, which allow us to improve the quality of the paper. We are very happy about your positive consideration of our manuscript. Below please find the response to your specific comments.

2. Specific comments

P9, L13 and eq. 5: It is not clear whether the N2 originating from non-labelled pools are considered similarly in the case of N2 because evidence of such N2 is described in section 3.2.2.

Unfortunately not. The method applied do not allow this because of the background unlabelled N2 present in the sample that cannot be distinguished from the N2 originating from unlabeled pool.

P9, eq. 6: In P8, the authors describe that three separated gas species (N2, N2+N2O, and N2O) were measured. Then why is fp_N2O not used in this equation? Or did they confirm mass balance like fp_N2+N2O = fp_N2 + fp_N2O?

The described calculation approach appeared to be more precise, because the quantification of fp_N2+N2O and fp_N2 base on the same method (similar peaks in the measurement procedure) and fp_N2O is determined from a different peak. Details are described in Lewicka-Szczebak et al., 2013, RCM. The balance was confirmed. This explanation will be added in the manuscript.

P11, L4: "average 15N abundance in nitrate" What was averaged? Initial and final values?

Yes. This will be clarified.

P18, section 3.2.2: Although the calculation procedure for r_N2O is explained in detail in section 2.5.2, results of r_N2O from the 15N treatment are not shown. Does this mean the 15N gas flux method failed to give r_N2O value?

These results are shown in the Fig. S2 together with the isotopomer results. We showed them there to allow for direct comparison with isotope data. But now we realise that this was a bad idea, and this should be shown in the graphs concerning 15N experiment (S3). This will be changed. This results are used all over the manuscript as a reference data for residual fraction for Exp.2.

P23, L6: "r_N2O values are always higher for Sc2" This is consistent with Figure 8, but I found the opposite statement in P16, L19. Please check the text and the figure.

True is: Sc 1 show lower rN2O, hence more N2O was reduced. P16, L19 must be corrected.

P23, L23: "for both soils in the anoxic treatment the cumulative non-labelled N2O flux is higher than the initial NH4+ pool plus the NH4+ possibly added" I could not follow. When I compare the 8th, 12nd, and 13th columns in Table S1, this is the case only for Min Soil with anoxic, 15N treatment.

This statement is true, but we poorly presented the units. The fluxes in Table S1 are given as rate for 24h, the unit should be described as $[mg\,N\,kg^{-1}\,d^{-1}]$

P24, L6: "it represents, respectively, 2 and 3 % of the nitrification rate (Table 1)" I cannot understand how I find this in Table 1. What does "respectively" means?

There is a mistake at the beginning of this paragraph – should be 'oxic treatments' instead of 'anoxic', which will be corrected. 'Respectively' means for Min and Org soil. It was calculated as: 0.01/0.3 = 0.033 and 0.07/1.93 = 0.036 – slight difference occurs due to rounding of the original values.

P24, L8: "observed increase in NO3-" In Table S1, C_NO3t is always lower than C_NO30, so there seems to be no "increase in NO3-". P27, L8: "15N-pool derived N2O characterized by higher d15Nsp values" In section 2.5, the authors did not mention that they measured d15Nsp of N2O in 15N gas flux method. Did they measure it?

This is based on the 15N dilution method. The concentration of nitrate decreases due to its consumption but based on 15N dilution we calculate how much nitrate was added to the nitrate pool. This will be clarified in the manuscript.

P32, L27: It seems that 15N gas flux method is useful to detect the processes such as producing hybrid N2O or N2, but I'm not convinced that it is really necessary to determine r_N2O (see above comment on section 3.2.2).

This was very useful and used in the manuscript as the main reference method for quantification of rN2O. We will add this data on the respective graphs (S3), and pay attention to make this clear in the manuscript. This information is very important and must be emphasised, we really regret that this was not sufficiently described.

3. Technical corrections

All the required corrections will be included in the manuscript. Thank you very much for your careful check.

**Anonymous Referee #3**

Referee comments in red,
authors response in black.

Thank you very much for reviewing our manuscript and for your valuable suggestions, which allow us to improve the quality of the paper. We are happy to hear you appreciate the scientific value of the paper and will do our best to improve the presentation of the material according to your suggestions.

The sentences were often long and awkwardly written.

We will rewrite the awkward sentences and improve the language quality of the paper.

Additionally, many of the terms were not well defined and described early in the manuscript. Then when the results were presented it was difficult to understand what each variable meant, and why they were important.

We will pay attention to precisely define each term in the manuscript.

The figures were also cluttered with too much information. I would have liked to seen there be more things distilled down for the reader, rather than showing all the data and every experiment. I wonder if all the experiments or data should be presented in one paper or if some of this information would be best split into multiple manuscripts.

We will try to simplify the figures, but we would not like to remove any data from the manuscript. We had been also thinking about it before to split the paper, but we decided, this makes not much sense, since the data are all connected and needed to check the performance of the reduction fractionation and mapping approaches (*i.e.,* isotope data to calculate SP and $d^{18}O$ of produced $N_2O$ and its uncertainty from Exp1 and Exp2, $^{15}N$ tracing data from parallel experiments – Exp2 - for independent estimates of $N_2O$ reduction as well as formation of hybrid $N_2$ and $N_2O$ to check for pathways other than bacterial denitrification, $N_2$ fluxes from He incubations – Exp1 - as the most precise independent measure of $N_2O$ reduction).

We believe it would be even more difficult for the reader if the data were split into separate papers. We wanted to present possibly all the basic data, because the methods we are describing are still under development. But all this very detailed information is placed in the supplement, and only readers especially interested in some particular points will need this. The paper should be understandable without this additional detailed information placed in the supplement. We will check the manuscript carefully, if this is the case.

Specific Comments:

P1 L1- I would suggest editing the title to make it catchier and less awkward.

Our new idea for the title is: Quantifying N2O reduction to N2 based on N2O isotopocules – validation with independent methods (Helium incubation and 15N gas flux method)

P1 L10- Rephrase, "the main unknown magnitude"

It will be rephrased to 'a missing quantity'.

P1 L11- Define in the abstract what the residual fraction is.

The definition will be added – remaining unreduced N2O.

P2 Introduction- Add more description on the importance of being able to determine N2O reduction. Also, give a better background on all the important terms to be later used in the manuscript and why they are important. A figure or table might be helpful for showing previous work and how the terms fit into the overall picture.

This information will be added: N2O reduction to N2 is the key quantity of N cycle that is poorly quantified, cause loss of fertilizer N and lowering of N leaching, and is the least well understood N flux. N2O reduction is crucial to know in order to close the nitrogen budget. We actually gave a background of the important terms discussed later, but we will try to better explain their importance.

P2 L4-6, P3 L1-3, P3 L26-28- Long and awkward sentences, consider rewriting.

They will be corrected as follows:

P2 L4-6 Commonly applied analytical techniques enable us to quantitatively analyse only the intermediate product of this process, $N_2O$, but not the final product, N2. This is due to the high atmospheric N2 background precluding direct measurements of $N_2$ emissions.

P3 L1-3 Its advantage over the [15]N gas flux method lies in its easier and non-invasive application, no need of additional fertilization, and much lower costs. This expands the application potential of the isotopic fractionation method and enables its more widespread use.

P3 L26-28 There are still some open questions: (i) whether the isotopic fractionation factors for denitrification processes determined in laboratory experiments are transferable to field conditions, (ii) how robustly the $N_2O$ residual fraction can be determined and (iii) whether the quantification of the entire nitrogen loss due to denitrification is possible.

P4 L19- The heading title has "experiment 1 and Exp 1" I would only write it once in the title.

This will be corrected, we wanted to introduce the abbreviation by the first mention of the particular experiment, and this is in this subtitle. But we will introduce this in the main text.

P5 L2- Missing "a" in "application of a N2-free atmosphere"

The mistake will be corrected.

This will be corrected.

P5 L21- Why is the nitrate treatment different in Exp2 than Exp1?

In Exp2 we applied more nitrate because this experiment lasted longer and the nitrate amendment was proportionally higher to ensure we have residual nitrate for analyses at the end of the experiment. We will add this explanation in the manuscript.

P6 L8- Define NA.

It was defined before, P5, L24.

Supplemental Figures S1 and S2- I suggest removing some of the variables from the figures and putting a simplified figure in the main text. I was also confused with the labeling and order of the figure S2, such that it went from 2.1 a) to 2.2 a) and then back to 2.1 b), could you combine panels onto 1 page and make them a b c and d?

The Figures will be improved and labelling will be corrected as suggested. We will delete the information about the residual N2O fraction (rN2O) from the graphs showing data from NA Experiments, since this information is gained from 15N Experiments and we realised it makes confusion. This will be shown in graphs for 15N experiments.
But we would not like to include this graphs in the paper main text, because this is lot of basic data, and they are all shown in other graphs in the paper, but not as time series. We believe the detailed time series are not the most important to show in the paper. This will make the reading rather more difficult.

P14 L24- In the "mapping approach" how much will the answer change if you use different end member values? The boxes for possible values are large and suspect it could be large.

Yes, this is a very important issue, but also quite complex, it will be addressed in the following paper, to be submitted very soon to Biogeosciences: *Buchen, C., Lewicka-Szczebak, D., Giesemann, A., Well, R., in preparation. Estimating N2O processes during grassland renovation and grassland conversion to arable land using N2O isotopocules*. In that manuscript we present the possible range of results taking into account the wide ranges of endmember values and also fractionation factors due to reduction. We will add the respective citation in this manuscript.

P15 L2-7: What is the value are you referring to in this paragraph?

To the values given above: from 17.4 to 21.4 ‰. We will clarify this.

P16 L1: I was surprised that the reduction isotope ratios were the same for oxic and anoxic incubations. Why is that?

The same process in both incubations is responsible for the N2O reduction. Oxic incubations were conducted in very humid conditions, hence, even for the oxic atmosphere many of soil microsites will maintain anoxic conditions.

P19 L3: What is N immobilization?

Transfer of mineral nitrogen into organic nitrogen forms. It has been defined earlier in the manuscript, in Section 2.6, Eq. (16).

P19 L17: What is hybrid N2O? Could you define it earlier in the manuscript?

It has been defined earlier in the manuscript, in Section 2.5.2, Eq. (10,11).

P20 L4: What correlations?

Correlations between N2O residual fraction $r_{N2O}$ and measured $\delta_r$ values. It was explained in 2.7.1, but we will repeat this information in the discussion.

P21-22: What are the differences between Val 1 and Val 2? Can you state them more clearly before presenting the results?

This was explained in the method section in 2.7.2, P14, L1-9.

P22 L25: Could the data in Table 3 be put into a simplified graph in the main text? That might be helpful for the reader.

We do not really see the possibility to present this information on a simple graph, because the table lists the results of 13 functions from several experiments. The graphs are presented in the supplement, and we will add this link to the heading.

P25 L10-15: I'd suggest putting the historical data in a table with the current findings.

We just wanted to indicate some typical problems and compare to our results. We will try to make the description more clear and add a table presenting 15N contents in soil nitrate and the released N2O in our and previous studies.

P27 L22: The title "Calibration and Validation" is vague, calibration and validation of what?

"-of rN2O quantification" - will be added to this subheading

Table 1: I suggest putting the full names of the variables in the table header row.

It will be corrected as suggested

Figure 2 and 3: Why is the x-axis on the right hand side?

No reason, we will move the y-axis on the left hand side.

Figure 4, 6 and 8: Symbols are similar and hard to distinguish in the figure.

We will use larger symbols.

Figure 5: There is no legend.

The legend will be added.

---

## Author Response (AR1)

*Review response on* "Isotopic fractionation of N2O to quantify N2O reduction to N2 – validation with Helium incubation and 15N gas flux methods" *by* Dominika Lewicka-Szczebak et al.

5  **Anonymous Referee #1**

Referee comments in red,
authors response in black,
corrections made in the manuscript in blue (given pages and lines of manuscript with tracked changes).

10  Thank you very much for reviewing our manuscript and for your valuable suggestions, which allowed us to improve the quality of the paper. We are very happy about your positive consideration of our manuscript. Below please find the corrections made in the manuscript according to your specific comments.

2. Specific comments

P9, L13 and eq. 5: It is not clear whether the N2 originating from non-labelled pools are considered similarly in the case of
15  N2 because evidence of such N2 is described in section 3.2.2.

Unfortunately not. The method applied do not allow this because of the background unlabelled N2 present in the sample that cannot be distinguished from the N2 originating from unlabeled pool.

P9, eq. 6: In P8, the authors describe that three separated gas species (N2, N2+N2O, and N2O) were measured. Then why is fp_N2O not used in this equation? Or did they confirm mass balance like fp_N2+N2O = fp_N2 + fp_N2O?

20  The described calculation approach appeared to be more precise, because the quantification of fp_N2+N2O and fp_N2 base on the same method (similar peaks in the measurement procedure) and fp_N2O is determined from a different peak. Details are described in Lewicka-Szczebak et al., 2013, RCM. The balance was confirmed.
This explanation has been added in the manuscript (P10,L8-13).

P11, L4: "average 15N abundance in nitrate" What was averaged? Initial and final values?

25  Yes. This has been clarified (P11,L17).

P18, section 3.2.2: Although the calculation procedure for r_N2O is explained in detail in section 2.5.2, results of r_N2O from the 15N treatment are not shown. Does this mean the 15N gas flux method failed to give r_N2O value?

These results were shown in the Fig. S2 together with the isotopomer results. We showed them there to allow for direct comparison with isotope data. But now we realised that this was a bad idea, and this should be shown in the graphs
5   concerning 15N experiment (S3). These results are used all over the manuscript as a reference data for residual fraction for Exp.2.

The figures S2 and S3 have been changed accordingly.

P23, L6: "r_N2O values are always higher for Sc2" This is consistent with Figure 8, but I found the opposite statement in P16, L19. Please check the text and the figure.

10  True is: Sc 1 show lower rN2O, hence more N2O was reduced.

P17, L20 has been corrected.

P23, L23: "for both soils in the anoxic treatment the cumulative non-labelled N2O flux is higher than the initial NH4+ pool plus the NH4+ possibly added" I could not follow. When I compare the 8th, 12nd, and 13th columns in Table S1, this is the case only for Min Soil with anoxic, 15N treatment.

15  This statement is true, but we poorly presented the units. The fluxes in Table S1 are given as rate for 24h, the unit should be described as [mg N kg$^{-1}$ d$^{-1}$].

The Table S1 has been corrected.

P24, L6: "it represents, respectively, 2 and 3 % of the nitrification rate (Table 1)" I cannot understand how I find this in Table 1. What does "respectively" means?

20  There is a mistake at the beginning of this paragraph – should be 'oxic treatments' instead of 'anoxic', which will be corrected. 'Respectively' means for Min and Org soil. It was calculated as: 0.01/0.3 = 0.033 and 0.07/1.93 = 0.036 – slight difference occurs due to rounding of the original values.

Anoxic has been corrected to oxic (P25,L22).

P24, L8: "observed increase in NO3-" In Table S1, C_NO3t is always lower than C_NO30, so there seems to be no "increase
25  in NO3-".

This is based on the 15N dilution method. The concentration of nitrate decreases due to its consumption but based on 15N dilution we calculate how much nitrate was added to the nitrate pool.

This has been changed in the manuscript to 'NO$_3^-$ production from nitrification' instead 'increase' (P25,L26).

P27, L8: "15N-pool derived N2O characterized by higher d15Nsp values" In section 2.5, the authors did not mention that they measured d15Nsp of N2O in 15N gas flux method. Did they measure it?

No, it was not measured – here the parallel experiments 15N and NA are interpreted jointly.
This has been clarified in the manuscript (P30, L12-14).

P32, L27: It seems that 15N gas flux method is useful to detect the processes such as producing hybrid N2O or N2, but I'm not convinced that it is really necessary to determine r_N2O (see above comment on section 3.2.2).

This was very useful and used in the manuscript as the main reference method for quantification of rN2O. This information is very important and must be emphasised, we really regret that this was not sufficiently described.
We have added this data on the respective graphs (S3), and paid attention to make this clear in the manuscript (more explanation added in 2.7.2, 3.2.1, 3.2.2).

3. Technical corrections

All the required corrections will be included in the manuscript. Thank you very much for your careful check.

**Anonymous Referee #3**

Referee comments in red,
authors response in black,
corrections made in the manuscript in blue (given pages and lines of manuscript with tracked changes).

Thank you very much for reviewing our manuscript and for your valuable suggestions, which allowed us to improve the quality of the paper. We are happy to hear you appreciate the scientific value of the paper and did our best to improve the presentation of the material according to your suggestions.

The sentences were often long and awkwardly written.

We have rewritten the awkward sentences and improved the language quality of the paper.

Additionally, many of the terms were not well defined and described early in the manuscript. Then when the results were presented it was difficult to understand what each variable meant, and why they were important.

We paid attention to precisely define each term in the manuscript.

The figures were also cluttered with too much information. I would have liked to seen there be more things distilled down for the reader, rather than showing all the data and every experiment. I wonder if all the experiments or data should be presented in one paper or if some of this information would be best split into multiple manuscripts.

We will try to simplify the figures, but we would not like to remove any data from the manuscript. We had been also thinking about it before to split the paper, but we decided, this makes not much sense, since the data are all connected and needed to check the performance of the reduction fractionation and mapping approaches (*i.e.,* isotope data to calculate SP and $d^{18}O$ of produced $N_2O$ and its uncertainty from Exp1 and Exp2, $^{15}N$ tracing data from parallel experiments – Exp2 - for independent estimates of $N_2O$ reduction as well as formation of hybrid $N_2$ and $N_2O$ to check for pathways other than bacterial denitrification, $N_2$ fluxes from He incubations – Exp1 - as the most precise independent measure of $N_2O$ reduction).

We believe it would be even more difficult for the reader if the data were split into separate papers. We wanted to present possibly all the basic data, because the methods we are describing are still under development. But all this very detailed information is placed in the supplement, and only readers especially interested in some particular points will need this. We have checked, that the paper should be understandable without this additional detailed information placed in the supplement.

Specific Comments:

P1 L1- I would suggest editing the title to make it catchier and less awkward.

Title has been changed to: Quantifying N2O reduction to N2 based on N2O isotopocules – validation with independent methods (Helium incubation and 15N gas flux method)

P1 L10- Rephrase, "the main unknown magnitude"

It has been rephrased to 'an important but rarely quantified process'. (P1, L12-13)

P1 L11- Define in the abstract what the residual fraction is.

The definition has been added – 'remaining unreduced N2O'. (P1, L14))

P2 Introduction- Add more description on the importance of being able to determine N2O reduction. Also, give a better background on all the important terms to be later used in the manuscript and why they are important. A figure or table might be helpful for showing previous work and how the terms fit into the overall picture.

5    This information has been added: N2O reduction to N2 is the key quantity of N cycle that is poorly quantified, cause loss of fertilizer N and lowering of N leaching, and is the least well understood N flux. N2O reduction is crucial to know in order to close the nitrogen budget. (P2,L12-16)

P2 L4-6, P3 L1-3, P3 L26-28- Long and awkward sentences, consider rewriting.

They have been corrected as follows:

10   P2 L6-8 Commonly applied analytical techniques enable us to quantitatively analyse only the intermediate product of this process, $N_2O$, but not the final product, N2. This is due to the high atmospheric N2 background precluding direct measurements of $N_2$ emissions.

P3 L12-15 Its advantage over the [15]N gas flux method lies in its easier and non-invasive application, no need of additional fertilization, and much lower costs. This expands the application potential of the isotopic fractionation method and enables
15   its more widespread use.

P4 L11-14 However, some open questions still remain: (i) are the isotopic fractionation factors for denitrification processes determined in laboratory experiments transferable to field conditions?; (ii) how robustly can the $N_2O$ residual fraction be determined?; (iii) is the quantification of the entire nitrogen loss due to denitrification possible?

20   P4 L19, P5 L15- The heading title has "experiment 1 and Exp 1" I would only write it once in the title.

We have introduced the abbreviation by the first mention of the particular experiment, and this is in this subtitle. We decided this will be strange and not so clear to define this abbreviation somewhere later.
Therefore, this is left when experiments are first mentioned, 2.1.1 and 2.1.2, and corrected in further paragraphs, 3.1 and 3.2

P5 L2- Missing "a" in "application of a N2-free atmosphere"

25   The mistake has been corrected.

P5 L21- Why is the nitrate treatment different in Exp2 than Exp1?

In Exp2 we applied more nitrate because this experiment lasted longer and the nitrate amendment was proportionally higher to ensure we have residual nitrate for analyses at the end of the experiment.

P6 L8- Define NA.

It was defined before, P5, L24.

Supplemental Figures S1 and S2- I suggest removing some of the variables from the figures and putting a simplified figure in the main text. I was also confused with the labeling and order of the figure S2, such that it went from 2.1 a) to 2.2 a) and then back to 2.1 b), could you combine panels onto 1 page and make them a b c and d?

We would not like to include this graphs in the paper main text, because this is lot of basic data, and they are all shown in other graphs in the paper, but not as time series. We believe the detailed time series are not the most important to show in the paper. This would make the reading rather more difficult.

The Figures S2 and S3 have been improved and labelling has been corrected as suggested. We have deleted the information about the residual N2O fraction (rN2O) from the graphs showing data from NA Experiments, since this information is gained from 15N Experiments and we realised it makes confusion. This has been shown in graphs for 15N experiments (S3).

P14 L24- In the "mapping approach" how much will the answer change if you use different end member values? The boxes for possible values are large and suspect it could be large.

Yes, this is a very important issue, but also quite complex, it will be addressed in the following paper, to be submitted very soon to Biogeosciences: *Buchen, C., Lewicka-Szczebak, D., Giesemann, A., Well, R., in preparation. Estimating N2O processes during grassland renovation and grassland conversion to arable land using N2O isotopocules.* In that manuscript we present the possible range of results taking into account the wide ranges of endmember values and also fractionation factors due to reduction.

We have added the respective explanation and citation in this manuscript. (P35, L26 – P36,L2)

P15 L2-7: What is the value are you referring to in this paragraph?

To the values given above: from 17.4 to 21.4 ‰.

It has been clarified. (P16, L4)

P16 L1: I was surprised that the reduction isotope ratios were the same for oxic and anoxic incubations. Why is that?

The same process in both incubations is responsible for the N2O reduction. Oxic incubations were conducted in very humid conditions, hence, even for the oxic atmosphere many of soil microsites will maintain anoxic conditions.

P19 L3: What is N immobilization?

Transfer of mineral nitrogen into organic nitrogen forms. It has been defined earlier in the manuscript, in Section 2.6, Eq. (16).

P19 L17: What is hybrid N2O? Could you define it earlier in the manuscript?

It has been defined earlier in the manuscript, in Section 2.5.2, Eq. (10,11).

P20 L4: What correlations?

Correlations between N2O residual fraction $r_{N2O}$ and measured $\delta_r$ values. It was explained in 2.7.1.
We have repeated this information in the discussion. (P21,L20)

P21-22: What are the differences between Val 1 and Val 2? Can you state them more clearly before presenting the results?

This was explained in the method section in 2.7.2, P14, L1-9.

P22 L25: Could the data in Table 3 be put into a simplified graph in the main text? That might be helpful for the reader.

We do not really see the possibility to present this information on a simple graph, because the table lists the results of 13 functions from several experiments. The graphs are presented in the supplement.
We have added this link to the table caption.

P25 L10-15: I'd suggest putting the historical data in a table with the current findings.

We have added a table presenting 15N contents in soil nitrate and the released N2 and N2O in our and previous studies (Table 4).

P27 L22: The title "Calibration and Validation" is vague, calibration and validation of what?

"-of rN2O quantification" - has been added to this subheading

Table 1: I suggest putting the full names of the variables in the table header row.

It has been corrected as suggested.

Figure 2 and 3: Why is the x-axis on the right hand side?

Because of the logarithmic scale of the x-axis, this axis cannot be started at 0, and all the very low values, at the border of detection limit are shown jointly at the beginning of x-axis as <0.01. The graphs will be less readable if we move the y-axis on the left hand side.

Figure 4, 6 and 8: Symbols are similar and hard to distinguish in the figure.

We have used larger symbols.

Figure 5: There is no legend.

The legend has been added.

[revised manuscript text omitted]

$$a_{\text{M\_N2O}} = a_{\text{P\_N2O}} \cdot f_{\text{P\_N2O}} + 0.003663 \cdot f_{\text{N\_N2O}} \tag{5}$$

where 0.003663 is the fraction of $^{15}$N in non-labelled N$_2$O and $f_{\text{N\_N2O}} = 1 - f_{\text{P\_N2O}}$.

Based on the determined $f_{\text{P\_N2}}$ and $f_{\text{P\_N2+N2O}}$ we can calculate $r_{\text{N2O}}$ as:

$$r_{\text{N2O}} = \frac{y_{\text{N2O}}}{y_{\text{N2}} + y_{\text{N2O}}} = \frac{f_{\text{P\_N2+N2O}} - f_{\text{P\_N2}}}{f_{\text{P\_N2+N2O}}} \tag{6}$$

where $y$ represents the mole fractions. This approach appeared to be more suitable than directly using $f_{\text{P\_N2O}}$, because (i) direct isotopic analysis of the N$_2$O was not possible in samples with low N$_2$O concentration and (ii) $f_{\text{P\_N2}}$ and $f_{\text{P\_N2+N2O}}$ were quantified in one sample based on the same method whereas $f_{\text{P\_N2O}}$ includes analysis of isotope ratios of the N$_2$O peak and analysis of N$_2$O conentration by gas chromatography in a replicate gas sample, thus resulting in potential bias in $f_{\text{P\_N2O}}$ due to the difficulty to collect exactly identical replicate gas samples. (Lewicka-Szczebak et al., 2013b).

Knowing $r_{\text{N2O}}$ we can estimate the total denitrification [N$_2$+N$_2$O] flux using the measured [N$_2$O] flux and the determined $r_{\text{N2O}}$ as:

$$[\text{N}_2 + \text{N}_2\text{O}]\,\text{flux} = \frac{[\text{N}_2\text{O}]\,\text{flux} \cdot f_{\text{P\_N2O}}}{r_{\text{N2O}}} + [\text{N}_2\text{O}]\,\text{flux} \cdot f_{\text{
[revised manuscript text omitted]